# AsCAN: Asymmetric Convolution-Attention Networks for Efficient Recognition and Generation

**Anil Kag**     **Huseyin Coskun**     **Jierun Chen**[*]    **Junli Cao**
**Willi Menapace**     **Aliaksandr Siarohin**     **Sergey Tulyakov**     **Jian Ren**
Snap Inc.
Project Page: https://snap-research.github.io/snap_image

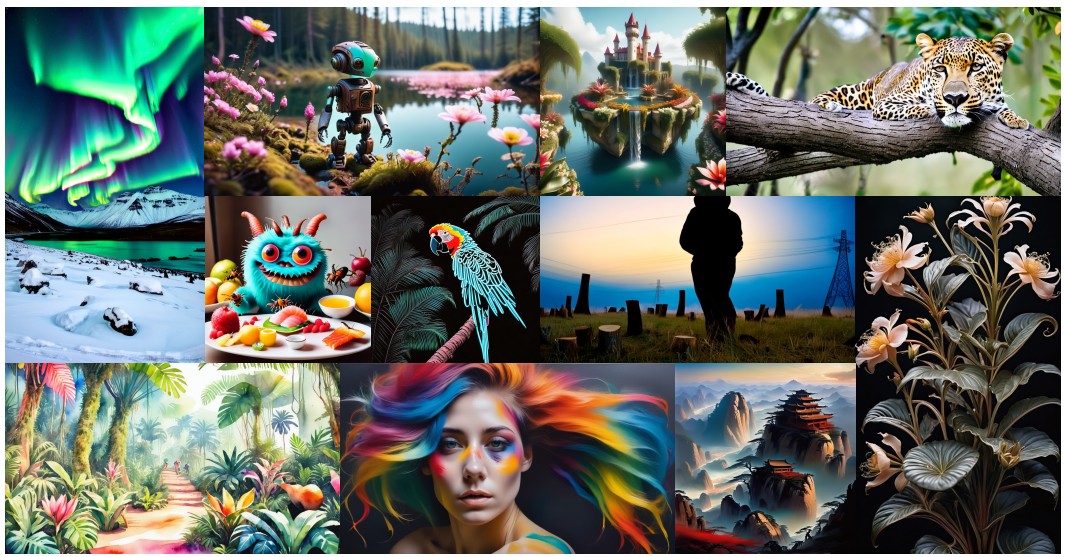

Figure 1: Example images generated by our efficient text-to-image generation model based on an asymmetric architecture. It generates photo-realistic images while following long prompts.

## Abstract

Neural network architecture design requires making many crucial decisions. The common desiderata is that similar decisions, with little modifications, can be reused in a variety of tasks and applications. To satisfy that, architectures must provide promising latency and performance trade-offs, support a variety of tasks, scale efficiently with respect to the amounts of data and compute, leverage available data from other tasks, and efficiently support various hardware. To this end, we introduce AsCAN—a hybrid architecture, combining both convolutional and transformer blocks. We revisit the key design principles of hybrid architectures and propose a simple and effective *asymmetric* architecture, where the distribution of convolutional and transformer blocks is *asymmetric*, containing more convolutional blocks in the earlier stages, followed by more transformer blocks in later stages. AsCAN supports a variety of tasks: recognition, segmentation, class-conditional image generation, and features a superior trade-off between performance and latency. We then scale the same architecture to solve a large-scale text-to-image task and show state-of-the-art performance compared to the most recent public

---

[*]Work done during an internship at Snap Inc.

38th Conference on Neural Information Processing Systems (NeurIPS 2024).

and commercial models. Notably, even without any computation optimization for transformer blocks, our models still yield faster inference speed than existing works featuring efficient attention mechanisms, highlighting the advantages and the value of our approach.

# 1 Introduction

Convolutional neural networks (CNNs) and transformers have been deployed in a wide spectrum of real-world applications, addressing various computer vision tasks, *e.g.*, image recognition [1, 2] and image generation [3, 4, 5, 6, 7, 8]. CNNs encode many desirable properties like translation equivariance facilitated through the convolutional operators [9]. However, they lack the input-adaptive weighting and the global receptive field capabilities offered by transformers [10, 11]. By recognizing the potential benefits of combining these complementary strengths, research endeavors explore *hybrid architectures* that integrate both convolutional and attention mechanisms [12, 13, 14, 15]. Recently, such architectures witness huge success when scaled up to train large-scale text-to-image (T2I) diffusion models [16, 17, 18, 19], enabling a vast range of visual applications, such as content editing [20, 21, 22, 23, 24, 25, 26] and video generation [27, 28, 29, 30].

One prominent research area for hybrid models involves the creation of building blocks that can effectively combine convolution and attention operators [31, 32]. While these efforts seek to use the strengths of both operators, their faster attention alternatives only approximate the global attention, leading to compromised model performance as lacking a global receptive field. Thus, they necessitate incorporating additional layers to compensate for the capacity reduction due to the attention approximation. On the other hand, minimal effort is directed toward optimizing the *entire* hybrid architecture. This raises the question: *Is the current macro design of hybrid architecture optimal?*

In this work, we propose a *simple* yet *effective* hybrid architecture, wherein the number of convolution and transformer blocks is *asymmetric* in different stages. Specifically, we adopt more convolutional blocks in the early stages, where the feature maps have relatively large spatial sizes, and more transformer blocks at the later stages. This design is verified across different tasks. For example, in Fig. 3, we demonstrate superior advantages of our model for the latency-performance trade-off on ImageNet-1K [33] classification task. Particularly, our model achieves even *faster speed* than many works featuring efficient attention operations. Additionally, we scale up our architecture to train the large-scale T2I diffusion model for high-fidelity generation (Fig. 1, Fig. 4, and Tab. 3). Furthermore, considering the high training cost for the large-scale T2I diffusion models, we introduce a multi-stage training pipeline to improve the training efficiency. Overall, our contributions can be summarized as follows:

- We revisit the macro design principles of hybrid convolutional-transformer architectures and propose one with asymmetrically distributed convolutional and transformer blocks.

- For the image classification task, we perform extensive latency analysis on the ImageNet-1K dataset and show our models achieve superior throughput-performance trade-offs than existing works (see Fig. 3). Notably, we show that the model runtime can be significantly accelerated *even without any acceleration optimization on attention operations*. Additionally, we show our pre-trained models on ImageNet-1K can be applied to downstream tasks such as semantic segmentation.

- For the class-conditional generation on ImageNet-1K ($256 \times 256$), our asymmetric UNet achieves similar performance as state-of-the-art models with half the compute resources (see Tab. 4).

- For the text-to-image generation task, we demonstrate that our network can be scaled up for the large-scale T2I generation with a better performance-latency trade-off than existing public models (as in Tab. 3). Additionally, we improve the training efficiency through a multi-stage training pipeline, where we first train the model on a small dataset, *i.e.*, ImageNet-1K, for T2I generation, then fine-tune the model on a large-scale dataset.

# 2 Related Works

**Efficient Hybrid Architectures.** Over the past decade, convolutional neural networks (CNNs) [34, 35, 36] have achieved unprecedented performance in various computer vision tasks [37, 38]. Despite numerous attempts to improve CNNs [39, 40, 41, 42, 43], they still face limitations, particularly in terms of their local and restrictive receptive fields. On the other hand, the vision transformer (ViT)

treats images as sequences of tokens [44], facilitating the computation of global dependencies between tokens to enhance the receptive field. However, ViTs come with quadratic computation complexity concerning input resolution. To address this, several studies have aimed to develop more efficient attention operations [45, 46, 47, 48, 49]. Recently, there has been a growing interest in exploring models that go beyond pure convolutional or transformer blocks, combining them within a single architecture to harness the spatial and translational priors from convolutions and the global receptive fields from the attention mechanism [11, 31, 2, 12, 32]. We revisit the design choices of hybrid architectures and propose a new design with a fast runtime while maintaining high performance. We reveal that optimizing the macro architecture directly allows us to use the original attention for a superior latency-performance trade-off compared to existing works. Most importantly, our designed networks can be applied to different domains, *e.g.*, image recognition and image generation, with both superior latency-performance trade-offs.

**Text-to-Image Diffusion Models.** The development of text-to-image models, such as GAN-based models [50, 7], autoregressive models [51, 52], and diffusion models [3, 53], has enabled the generation of high-fidelity images by using textual descriptions. Among them, diffusion models demonstrate advantages in stable training processes and the scalability of large-scale neural networks [19, 54, 55, 16, 17, 13]. Recent studies explore the enhancements of text-to-image diffusion models through various directions, such as designing new network architectures [56, 57, 58, 59, 60, 61, 62, 63], improving inference efficiency during multi-steps sampling [64, 65, 66], and accelerating the training convergence [67, 68, 69, 70, 71]. Our designed network can be scaled up for the T2I generation when modified to a UNet architecture [13]. The model achieves better performance-latency trade-offs than open-sourced models. Furthermore, we can reduce the training cost through the proposed two-stage training pipeline.

## 3 Methods

This section describes our network and training designs in detail. First, we justify our choices of utilizing the convolutional and transformer operations for the building blocks, which we then use to design the asymmetric architecture (Sec. 3.1). Second, we scale up the network architecture for training the text-to-image diffusion models (Sec. 3.3). We point out that it is easier to ablate the design of building blocks on the image classification task due to the lower resource requirements (training/evaluation compute) in comparison to image generation.

### 3.1 Architecture Design: Asymmetric Convolution-Attention Networks (AsCAN)

Our hybrid architecture consists of a mix of convolutional and transformer blocks, which enables operating on different input resolutions seamlessly and allows us to be pareto efficient *w.r.t.* performance-compute trade-off compared to other architectures. Before delving into the exact architecture configuration, we discuss our choice of the building blocks. We use $X \in \mathbb{R}^{H \times W \times C}$ to represent the input feature map $X$ that has $H \times W$ spatial dimensions along with $C$ channels. We denote $Y \in \mathbb{R}^{H^{'} \times W^{'} \times C^{'}}$ as the output of a building block (convolution or transformer) and $\circ$ symbol for the function composition operator. In the following, we present our design choices based on ImageNet-1K [72] classification task by varying different convolution and transformer blocks.

**Convolutional Block (C).** There are various choices for a convolutional block that can be used in our architecture, *e.g.*, MBConv [73], FusedMBConv [1], and ConvNeXt [74]. While MBConv block has been used in many networks [75, 31], the presence of depthwise convolutions results in low accelerator utilization for high-end GPUs [1]. To better understand the latency-performance trade-off for various convolutional blocks, we experiment with the *same* hybrid architecture yet with *different* convolutional blocks. As in Tab. 1 (more experimental settings in Sec. 4.1), FusedMBConv has better latency on A100 and V100 GPUs than others, while maintaining high performance. Therefore, we adopt FusedMBConv (C) for the convolutional block and represent it as $Y = X + \mathcal{P} \circ \mathcal{SE} \circ \mathcal{C}(X)$, where $\mathcal{C}$ is a full $3 \times 3$ convolution with $4\times$ channel expansion followed by batch norm and GeLU non-linearity [76], $\mathcal{SE}$ is a squeeze-and-excite operator [39] with shrink ratio of $0.25$, and $\mathcal{P}$ is a $1 \times 1$ convolution to project to $C$ channels.

**Transformer Block (T).** Similar to the convolution blocks, there are many choices for attention mechanism in transformers, *e.g.*, blocks containing efficient attention mechanisms like multi-axial attention [31] and hierarchical attention [32]. Tab. 2 shows that vanilla attention mechanism provides

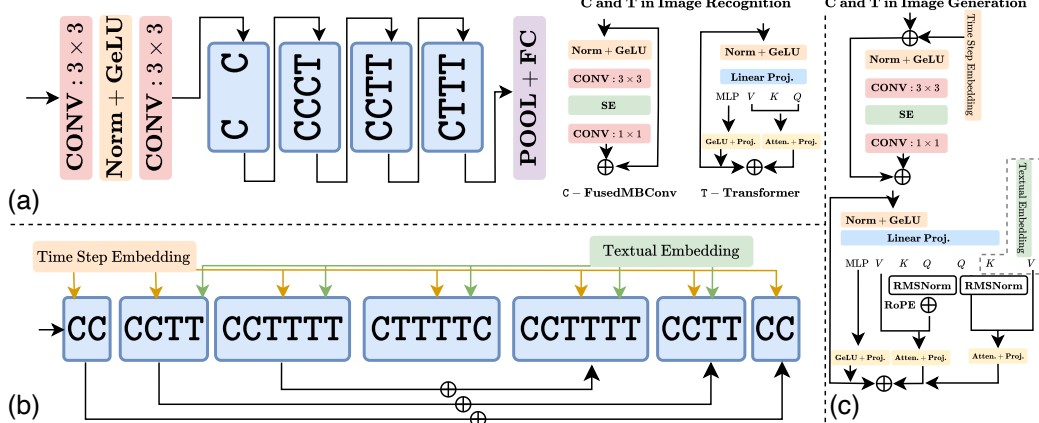

Figure 2: **Example AsCAN architectures for Image Classification & Text-to-Image Generation.** **(a):** The architecture for the image classification and details of the convolutional (`C`) and transformer blocks (`T`). AsCAN includes Stem (consisting of convolutional layers) and four stages followed by pooling and classifier. **(b):** The UNet architecture for the image generation. The Down blocks (the first three blocks starting from *left*) have the reverted reflection as the Up blocks (the first three blocks starting from *right*). **(c):** The details for `C` and `T` used in UNet. For the `T` that performs the cross attention between latent image features and textural embedding, the $Q$ matrix comes from the textural embedding. Note that, compared to image classification, the `C` and `T` blocks for image generation only adds extra components to incorporate the input time-step and textual embeddings.

a better accuracy *vs* throughput trade-off across different GPUs and batch sizes. Thus, we choose the vanilla attention in our transformer block (`T`). We express it with the following update equations:

$$Y = X + Y_{\text{attn}} + Y_{\text{mlp}}; \ Y_{\text{attn}} = \mathcal{P} \circ \mathcal{A} \circ \mathcal{P}_{\text{QKV}}(\hat{X}); \ Y_{\text{mlp}} = \mathcal{P} \circ \phi \circ \mathcal{P}_{\text{MLP}}(\hat{X}), \tag{1}$$

where $\hat{X} = \mathcal{LN} \circ \phi(X)$, $\mathcal{LN}$ denotes layer normalization, $\phi$ denotes the GeLU non-linearity [76], $\mathcal{A}$ is the multi-headed self-attention function, $\mathcal{P}_{\text{QKV}}$ & $\mathcal{P}_{\text{MLP}}$ denote the linear projection to the QKV and MLP space, respectively, and $\mathcal{P}$ denotes the projection operator to the same space as the input. Note that these update equations are inspired by recent works [77, 78], where the feed-forward and the self-attention operators are arranged in parallel in order to get improved throughput with marginal reduction in performance.

**Design Choices.** Given the FusedMBConv (`C`) and Vanilla Transformer (`T`) blocks, we introduce the macro design for our hybrid architecture. For the image classification task, we follow the existing works [11, 31, 32] to utilize a four-stage architecture (excluding the convolutional Stem at the beginning and classifier components at the end). However, before finalizing our architecture, we still have a design question to answer, namely, *in which configuration should we arrange these building blocks?* For instance, CoAtNet [11] chooses to stack convolutional blocks in the first two stages and transformer blocks in the remaining stages while MaxViT [31] stacks convolutional and transformer blocks alternatively throughout the entire network. A formal algorithm requires evaluating all possible `C` and `T` configurations, which is computationally expensive. Even neural architecture search leads to exponential search space. Thus, we follow a naive strategy that is based on the following principles:

- `C` **before** `T`. In any stage, we prefer convolutions followed by transformer blocks to capture the global dependence between the features aggregated by the convolutions, as they can capture scale and translation-aware information. Our ablations in Appendix Tab. 9 justify this design choice.
- **Fixed first stage.** As transformer blocks have quadratic computation complexity in terms of the sequence length, we prefer the first stage to contain only convolutional blocks to improve the inference latency.
- **Equal blocks in remaining stages.** For ease of analysis, we fix the number of blocks in the remaining stages, *i.e.*, stages 2 to 4, to be four. Once we finalize the basic configuration, we can scale these stages similar to earlier works [31] to achieve larger models.
- **Asymmetric *vs* Symmetric.** We refer to the architectures as symmetric whenever `C` and `T` blocks are distributed equally within a stage. For example, a configuration of `CCCC-CCTT-TTTT` is

symmetric since both `C` and `T` blocks are equal within a stage. In contrast, the configuration of `CCCT-CCTT-CTTT` is asymmetric since `C` and `T` blocks are not equal in stages 2 and 4.

**Final Architecture.** Given these design principles, we list various promising configurations for the building blocks (`C` & `T`), and analyze their inference throughput and top-1 accuracy. Tab. 2 provides these configurations along with performance and runtime on different GPUs. For a better comparison with existing works, we also add the configurations of CoAtNet and MaxViT for reference. From the results, we can draw the following conclusions:

- Compared to symmetric architecture design, asymmetric distribution of `C` & `T` blocks yield a better trade-off for throughput and accuracy, as compared by configurations C1-C5 *vs* C6-C10.
- Higher number of transformer blocks in the early stages result in lower throughput, which can be observed by comparing the latency for the configurations of C2 *vs* C10, and C8 *vs* C9.
- While increasing the number of transformer blocks in the network improves the throughput, it does not result in improved accuracy, as demonstrated by C6 *vs* C9.

Given above analysis, we prefer the C1 configuration for simplicity along with better accuracy *vs* latency trade-off. Since transformer blocks capture global dependencies, we prefer at least some blocks in the early layers as well in conjunction with the convolutional blocks. Similarly, we prefer having few convolutional blocks in the later stages to capture the spatial, translation, or scale aware features. To be concrete, our final architecture includes a convolutional stem followed by four stages and the classifier head. In the first stage, we only keep convolutional blocks. In the second stage, we keep 75% convolutional and 25% transformer blocks. This trend is reversed in the final stage. For the third stage, we keep equal number of convolutional and transformer blocks. We visualize this configuration in Fig. 2 along with the diagrams representing the convolutional and transformer blocks.

**Remarks.** For simplicity, in this work, we focus on the configuration in which to combine the `C` & `T` blocks, and leverage vanilla quadratic attention and convolutional mechanisms. We can incorporate faster alternatives to quadratic attention to further boost the performance and latency trade-offs. We leave this exploration to future research.

## 3.2 Discussion

While many works in the literature focus on improving the trade-off between performance and multiply-add operation counts (MACs), most of the time, MACs do not translate to throughput gains. It is primarily due to the following reasons:

- *Excessive use of operations that do not contribute to MACs.* Such tensor operators include reshape, permute, concatenate, stack, etc. While these operations do not increase MACs, they burden the accelerator with tensor rearrangement. The cost of such rearrangement grows with the size of the feature maps. Thus, whenever these operations occur frequently, the throughput gains drop significantly. For instance, MaxViT [31] uses axial attention that includes many permute operations for window/grid partitioning of the spatial features. Similarly, SMT [79] includes many concatenation and reshape operations in the SMT-Block. It reduces the throughput significantly even though their MACs are lower than AsCAN (see Appendix Tab. 7).
- *MACs do not account for non-linear accelerator behavior in batched inference.* Another issue is that MACs do not account for the non-linear behavior of the GPU accelerators in the presence of larger batch sizes. For instance, with small batch sizes (B=1), the GPU accelerator is not fully utilized. Thus, the benchmark at this batch size is not enough. Instead, one should benchmark at larger batch sizes to see consistency between architectures.
- *Lack of efficient CUDA operators for specialized building blocks.* Many architectures propose specialized and complex attention or convolution building blocks. While these blocks offer better MACs-vs-performance trade-offs, it is likely that their implementation relies on naive CUDA constructs and does not result in significant throughput gains. For instance, Bi-Former [80] computes attention between top-k close regions using a top-k sorting operation and performs many gather operations on the queries and keys. Similarly, RMT [81] computes the Manhattan distance between the tokens in the image. It includes two separate attention along the height and width of the image. This process invokes many small kernels along with reshape and permute operations. These specialized blocks would benefit from efficient CUDA kernels.
- *Using accelerator-friendly operators.* Depending on the hardware, some operators are better than others. Depth-wise separable convolutions reduce the MACs, yet they may not be efficient for

Table 1: **Analysis of the Configuration of Convolution and Transformer Blocks.** We analyze various convolutional and transformer blocks by training hybrid architectures with different options on the ImageNet-1K dataset. We provide the inference latency (as throughput) for different GPUs. Results show that FusedMB-Conv and Vanilla Transformer blocks provide a better trade-off over accuracy and latency than others.

| Convolution-Based Block | Params | (Images/s) A100 B=64 | V100 B=16 | Top-1 Acc. |
|---|---|---|---|---|
| MBConv | 29M | 3013 | 914 | 83.12% |
| ConvNext | 35M | 3923 | 1104 | 82.81% |
| FusedMBConv | 55M | 4295 | 1148 | 83.44% |

| Transformer-Based Block | Params | (Images/s) A100 B=64 | V100 B=16 | Top-1 Acc. |
|---|---|---|---|---|
| Multi-Axial | 83M | 3541 | 630 | 83.59% |
| Hierarchical | 74M | 3470 | 552 | 83.51% |
| Vanilla | 55M | 4295 | 1148 | 83.44% |

Table 2: **Analysis of Architecture Configuration.** We analyze the distribution of convolution and transformer blocks by training hybrid architecture with different distributions of blocks on ImageNet-1K dataset [72]. It demonstrates that the design in Fig. 2 provides a better trade-off over accuracy and latency. Symbol M denotes MaxViT block [31] composed of MBConv[73, 75] and Multi-Axial Attention blocks, which is equivalent to CT. Note that CoAtNet [11] uses MBConv blocks compared to the FusedMBConv blocks in our design.

| Family | Block Configuration | Params | Inference(images/s) A100 B=16 | A100 B=64 | V100 B=16 | Top-1 Acc. |
|---|---|---|---|---|---|---|
| Our Asymmetric | CC-CCCT-CCTT-CTTT (C1) | 55M | 3224 | 4295 | 1148 | 83.4% |
| | CC-CCCT-CCTT-CCTT (C2) | 73M | 3217 | 4179 | 1036 | 83.2% |
| | CC-CCCT-CCTT-TTTT (C3) | 41M | 3384 | 4472 | 1224 | 82.9% |
| | CC-CCCT-CCCC-TTTT (C4) | 50M | 3434 | 4411 | 1182 | 83.1% |
| | CC-CCCT-CCCT-CCCT (C5) | 95M | 3135 | 4066 | 991 | 82.7% |
| Our Symmetric | CC-CCCC-CCTT-TTTT (C6) | 51M | 3783 | 4998 | 1280 | 82.8% |
| | CC-CCCC-CCTT-TTTT (C7) | 42M | 3536 | 4941 | 1296 | 82.4% |
| | CC-CCCC-TTTT-TTTT (C8) | 34M | 3475 | 5311 | 1469 | 82.6% |
| | CC-TTTT-TTTT-TTTT (C9) | 30M | 3216 | 4091 | 1293 | 82.7% |
| | CC-CCTT-CCTT-CCTT (C10) | 72M | 2942 | 3820 | 980 | 82.8% |
| CoAtNet-0 | CC-CCC-TTTTT-TT | 25M | 3537 | 5221 | 976 | 81.6% |
| CoAtNet-1 | CC-CCCCCC-TTTTTTTTTTTTTTT-TT | 42M | 2221 | 2907 | 629 | 83.3% |
| MaxViT-T | MM-MM-MMMMM-MM | 31M | 1098 | 2756 | 357 | 83.6% |

particular hardware. Excessive use of depth-wise separable convolutions should be avoided in favor of the full convolutions wherever possible. For instance, MogaNet [82] extensively uses depth-wise convolutions with large kernel sizes and concatenation operations. These operators reduce the multiply-addition counts, which are not necessarily efficient on high-end GPU accelerators.

### 3.3 Scaling Up Architecture for Image Generation

We further scale up our architecture for the image generation task, which requires more computation than the image recognition task. We train our network by utilizing the denoising diffusion probabilistic models (DDPM) [16, 17, 54] in the latent space [13]. The latent diffusion model includes a variational autoencoder (VAE) [83, 84] that encodes the image $\mathbf{x}$ into latent $\mathbf{z}$ and a diffusion model $\hat{\epsilon}_{\theta}(\cdot)$ with parameters $\theta$. We utilize UNet [18] as the network for the diffusion model. For the T2I generation, we get the text embedding with Flan-T5-XXL encoder [85]. We train the diffusion model following the noise prediction [16, 17]:

$$\min_{\theta} \mathbb{E}_{t\sim[1,T],\epsilon\sim\mathcal{N}(\mathbf{0},\mathbf{I})}\|\epsilon - \hat{\epsilon}_{\theta}(\mathbf{z}_t, \mathbf{c})\|^2, \tag{2}$$

where $t$ is the time step and $T$ is the total number of steps, $\epsilon$ is the added Gaussian noise, and $\mathbf{c}$ is the condition signal. $\mathbf{z}_t$ is a noisy latent obtained with a pre-defined variance schedule $\{\beta_t \in (0, 1)\}_t$ with the following updates [16]:

$$\mathbf{z}_t = \sqrt{\bar{\alpha}_t}\mathbf{z} + \sqrt{1 - \bar{\alpha}_t}\epsilon; \quad \alpha_t = 1 - \beta_t; \quad \bar{\alpha}_t = \prod_{i=1}^{t}\alpha_i. \tag{3}$$

In the following, we discuss our design for the diffusion model and the multi-stage training pipeline.

#### 3.3.1 Asymmetric UNet Architecture

We follow the existing literature [13, 8, 15, 86] to design the UNet architecture for the T2I diffusion model. Fig. 2 gives an overview of the overall architecture. It consists of three main stages, namely, Down blocks, Middle Block, and Up blocks. The Up blocks are the reverted reflection of the Down blocks. In addition, there are skip connections to add the features from Down blocks to Up blocks. We adopt the VAE from SDXL [8] to transform the image to the latent space and carry out the diffusion in this space. Since the text-to-image generation task requires additional inputs (*i.e.*, time and text embeddings), we modify the C and T blocks to incorporate these conditions. Similar to the existing literature [8], we add the time embedding to the C blocks. Additionally, we add the text embedding to the T blocks through a cross-attention operation carried in parallel to the self-attention operation as shown in Fig. 2(c).

While the UNet can take arbitrary resolution as input, we notice that adding positional embeddings in the self-attention operation reduces the artifacts such as duplicate patterns. For this purpose, we add RoPE [87] embeddings for encoding the position information in the T blocks. Further, we incorporate query-key normalization using the RMSNorm [88] for stable training in lower precision (bfloat16).

### 3.3.2 Improved Multi-Stage Training Pipeline

Instead of training the T2I network from scratch, we first train the model on a small-scale dataset and then fine-tune the model on a much larger dataset. It effectively reduces the training cost on the large-scale scale dataset (see ablation in Appendix Sec. A.6 , Fig 8). This strategy differs from existing works that perform multi-stage training using different architectures. For instance, PixArt-$\alpha$ [69, 71] trains a class conditional image generation network and modifies it to fine-tune the network for text-to-image generation. Both our pre-training and fine-tuning tasks use the same architecture for text-to-image generation. We use the AdamW optimizer [89] with $\beta_1 = 0.9$ and $\beta_2 = 0.99$.

Specifically, in the first stage, we train our model using ImageNet-1K [33] for the text-to-image generation. Following Esser *et al.* [90], we form the conditioned text prompt as "a photo of a <class name>", where class name is randomly chosen from the label of each image. The model is trained to generate an image with a resolution of $256 \times 256$. In the second stage, we fine-tune the model from the first stage on a much larger dataset. Here we train the model in four phases: First, we conduct training at the resolution of $256 \times 256$ for 300K iterations with the batch size as $16, 384$ and $4e - 4$ as the learning rate. Second, we continue the training at the resolution of $512 \times 512$ for 200K iterations with the batch size as $6, 144$ and $1e - 4$ as the learning rate. Third, we train the model for $1024 \times 1024$ for 100K iterations with the batch size as $1, 536$ and $5e - 5$ as the learning rate. Finally, we perform the multi-aspect ratio training such that the model can synthesize images at various resolution [8]. Additionally, we adjust the added noise at different resolution [15], *i.e.*, $\beta_T$ in Eq. (3) is chosen as $0.01$ for $256 \times 256$ and $0.02$ for higher resolution (see ablations in Appendix Sec. A.6). We add an offset noise of $0.05$ during multi-aspect ratio training similar to earlier works[8].

## 4 Experiments

In this section, we evaluate the architectures proposed in Sec. 3.1 on image recognition and generation tasks. We also apply this design to the semantic segmentation task in Appendix Sec. A.3. We highlight the important aspects of these experiments below and provide the experimental details in the appendix.

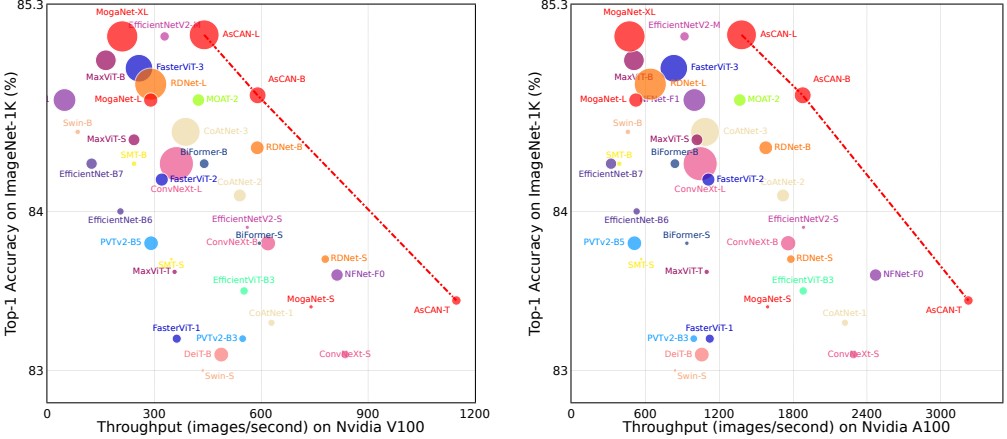

Figure 3: **Top-1 Accuracy *vs* Inference Latency on ImageNet-1K Classification.** We plot the latency measured as images inferred per second on a single V100 GPU (*Left*)/A100 GPU (*Right*) with batch-size 16 with $224 \times 224$ resolution. The plot compares state-of-the-art models (convolutional, transformer, hybrid architectures) against the proposed AsCAN architecture. The area of each circle is proportional to the model size. Our model consistently achieves better accuracy *vs* latency trade-offs. While some models regress between two hardware (*e.g.*, MaxViT-S vs SMT-B ), our model consistently achieves better accuracy *vs* latency trade-offs. We report additional baselines along with multiply-add operations count and different batch sizes in Appendix Tab. 7.

## 4.1 Image Recognition

We scale the AsCAN architecture discussed in Sec. 3.1 to base and large variants following earlier works [31]. We train these variants on the ImageNet-1K[72] classification task. We provide the experimental details (dataset description, architectures, training hyper-parameters) in Appendix Sec. A.2. Fig. 3 plots the inference speed on a V100 GPU with batch size 16 (measured in images processed per second) and top-1 accuracy achieved on this task for various models. In addition, Appendix Tab. 7 shows the parameter count and inference speed on both V100 and A100 GPUs along with additional details such as floating-point multiply-add count (FLOPs/MACs) and inference speed across different batch sizes. Below, we highlight the salient observations from these experiments.

- *More than $2\times$ higher throughput across accelerators.* Compared to the existing hybrid architectures such as FasterViT [32] and MaxViT [31], the proposed AsCAN family achieves similar or better top-1 accuracy with more than $2\times$ higher throughput. This trend holds true for both A100 and V100 GPUs. For instance, on A100 with batch=16, FasterViT-1 achieves 83.2% top-1 accuracy with throughput as 1123 images/s, while AsCAN-T has 83.44% with throughput as 3224 images/s.
- *Better throughput across different batch sizes.* AsCAN consistently achieves better throughput across batch sizes for both the accelerators compared to baselines.
- *Better storage and computational footprint.* AsCAN-family of architectures requires less number of parameters and float operations to achieve similar performance. For example, to achieve nearly 85.2% accuracy, MaxViT-L requires 212M parameters and 43.9G MACs whereas AsCAN-L requires 173M parameters and 30.7G MACs.
- We achieve better latency *vs.* accuracy trade-off than hybrid architectures with better alternatives to the attention mechanisms in transformer. For instance, we outperform newer architectures such as PVTv2[91], MOAT[92], EfficientViT[93], and Scale-Aware Modulation Transformers[79].

Further, we observe similar trends when we scale these models to pre-training on *ImageNet-21K* [94] dataset (see Appendix Sec. A.2.3) as well as semantic *segmentation* task (Appendix A.3).

Table 3: **GenEval Scores.** We use this benchmark to compare T2I models on various aspects of generation, including counting, color attribution, position, etc. It clearly shows that our model achieves better overall score, by convincingly outperforming pipelines such as PixArt-$\alpha$ and SDXL.

| Model | Overall | Single Object | Two Objects | Counting | Colors | Position | Color Attribution |
|---|---|---|---|---|---|---|---|
| SDv1.5 | 0.43 | 0.97 | 0.38 | 0.38 | 0.76 | 0.04 | 0.06 |
| PixArt-$\alpha$ | 0.48 | 0.98 | 0.50 | 0.44 | 0.80 | 0.08 | 0.07 |
| SDv2.1 | 0.50 | 0.98 | 0.51 | 0.44 | 0.85 | 0.07 | 0.17 |
| PixArt-$\Sigma$ | 0.53 | 0.99 | 0.65 | 0.46 | 0.82 | 0.12 | 0.12 |
| SDXL | 0.55 | 0.98 | 0.74 | 0.39 | 0.85 | 0.15 | 0.23 |
| Ours | 0.64 | 0.99 | 0.78 | 0.43 | 0.88 | 0.28 | 0.48 |

## 4.2 Class Conditional Generation

We apply our asymmetric UNet architecture to learn class conditional image generation on the ImageNet-1K dataset with $256 \times 256$ resolution. We train a smaller variant of our asymmetric UNet architecture (as in Sec. 3.3.1) with nearly 400M parameters and inject the class condition through cross-attention mechanism. We train this model using DDPM [16] (more details in Appendix Sec. A.7), and provide results in Tab. 4. As can be seen, we achieve Fréchet Inception Distance (FID) [95] close to state-of-the-art models with nearly half the FLOPs (*e.g.*, Ours *vs.* DiT-XL/2-G), and better FID than the existing work with similar computation (*e.g.*, Ours *vs.* U-ViT-L/2).

## 4.3 Text-to-Image Generation

Below, we evaluate our T2I asymmetric UNet architecture trained using the multi-stage training.

**GenEval Benchmark.** We evaluate our model on the GenEval [97] benchmark that studies various aspects of an image generation model. Tab. 3 shows that our model outperforms fast training pipelines such as PixArt-$\alpha$ by a significant margin. It beats even larger models such as SDXL which consume significantly more training data and computational resources.

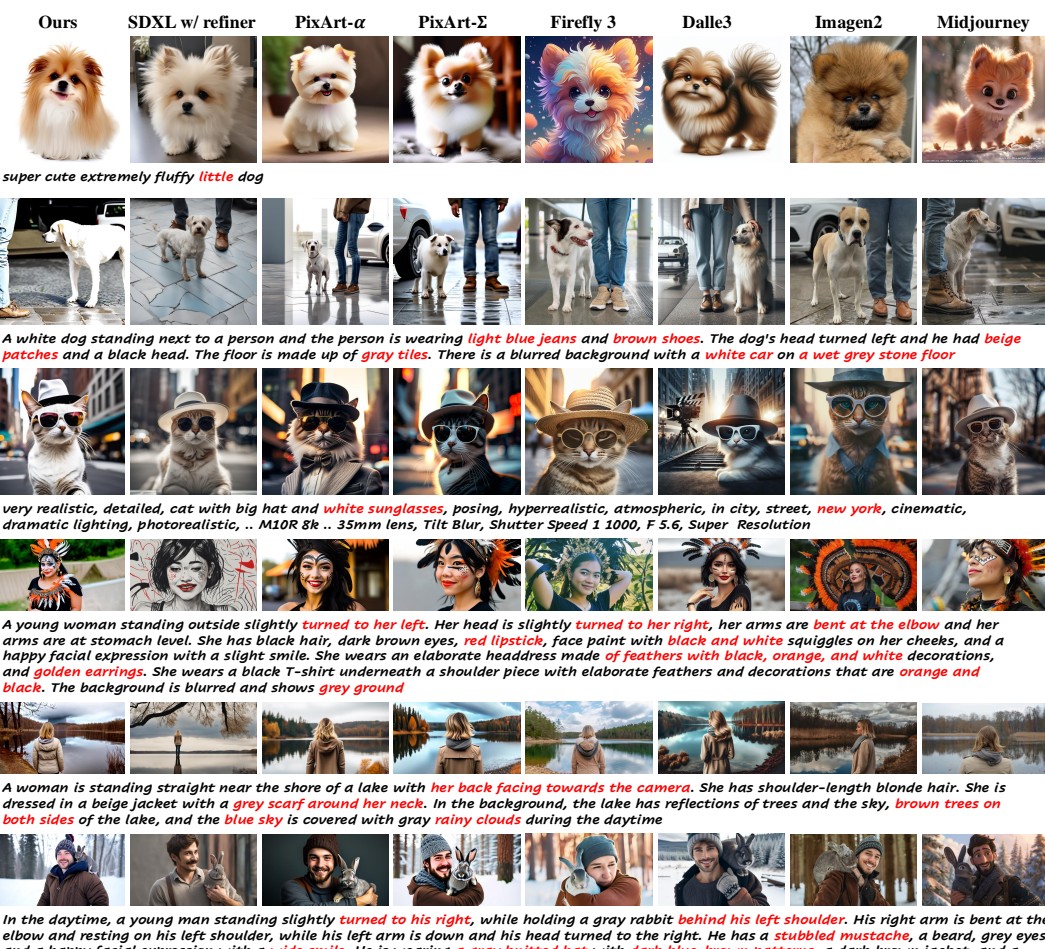

| Ours | SDXL w/ refiner | PixArt-α | PixArt-Σ | Firefly 3 | Dalle3 | Imagen2 | Midjourney |

*super cute extremely fluffy **little** dog*

*A white dog standing next to a person and the person is wearing **light blue jeans** and **brown shoes**. The dog's head turned left and he had **beige patches** and a black head. The floor is made up of **gray tiles**. There is a blurred background with a **white car** on **a wet grey stone floor***

*very realistic, detailed, cat with big hat and **white sunglasses**, posing, hyperrealistic, atmospheric, in city, street, **new york**, cinematic, dramatic lighting, photorealistic, .. M10R 8k .. 35mm lens, Tilt Blur, Shutter Speed 1 1000, F 5.6, Super Resolution*

*A young woman standing outside slightly **turned to her left**. Her head is slightly **turned to her right**, her arms are **bent at the elbow** and her arms are at stomach level. She has black hair, dark brown eyes, **red lipstick**, face paint with **black and white** squiggles on her cheeks, and a happy facial expression with a slight smile. She wears an elaborate headdress made **of feathers with black, orange, and white** decorations, and **golden earrings**. She wears a black T-shirt underneath a shoulder piece with elaborate feathers and decorations that are **orange and black**. The background is blurred and shows **grey ground***

*A woman is standing straight near the shore of a lake with **her back facing towards the camera**. She has shoulder-length blonde hair. She is dressed in a beige jacket with a **grey scarf around her neck**. In the background, the lake has reflections of trees and the sky, **brown trees on both sides** of the lake, and the **blue sky** is covered with gray **rainy clouds** during the daytime*

*In the daytime, a young man standing slightly **turned to his right**, while holding a gray rabbit **behind his left shoulder**. His right arm is bent at the elbow and resting on his left shoulder, while his left arm is down and his head turned to the right. He has a **stubbled mustache**, a beard, grey eyes and a happy facial expression with a **wide smile**. He is wearing a **gray knitted hat** with **dark blue-brown patterns**, a dark brown jacket, and a black sweater. There is a snowy forest with snow-capped trees in the background.*

Figure 4: **Qualitative Comparison against open source and commercial models.** We compare our T2I model against generations from different baselines. We illustrate that many times existing models generate images with less photo-realism (either lot less details or more on the cartoonish side), specially for PixArt-α and PixArt-Σ. Further, they frequently miss the fine-grained details explicitly asked in the prompts. We highlight these mistakes in red color in the input prompt. For instance, in the above generations (ordered A → F from top to bottom row), baselines miss details such as, (A) lack of realism (B) light blue jeans, (C) white sunglasses, (D) black, orange, and white feathers, (E) grey scarf & back towards camera, and (F) gray knitted hat with dark blue-brown patterns.

**Qualitative Comparison.** Fig. 4 shows a qualitative comparison between different methods. We highlight salient differences between the baselines and our generations. Baselines such as PixArt-α and PixArt-Σ tend to generate images with much less photo-realism and more often it is much more on the cartoonish side or it has grainy artifacts. Similarly, SDXL with refiner framework is unable to adapt to long text prompts due to limitations of the CLIP text-encoder. Hence, it misses many key features described in the prompts. In contrast, our model is able to follow the long prompts while adhering to the required semantics as well as photo-realism.

**Human Preference study.** We perform a user study to compare open-sourced models to evaluate their image-text alignment characteristics. We select 1000 prompts from our validation set and generate images from SDXL, PixArt-α, PixArt-Σ, and our multi-aspect ratio model. We show these results in the Fig. 5. It shows that our model convincingly outperforms both SDXL and PixArt-α generations, while it has similar image-text alignment as PixArt-Σ model. To further evaluate these models, we generate 10K images from our validation set (described in Appendix A.5.2) and compute the FID [95] between the generated and original images. We show these scores in Tab. 5, which

Table 4: **ImageNet-1K Class Conditional Generation.** We train a smaller variant of our T2I UNet architecture to perform class conditional generation on ImageNet-1K. We train this model at $256 \times 256$ and $512 \times 512$. Our asymmetric architecture achieves similar FID as the state-of-the-art models with less than half the floating point operations (*e.g.*, Ours *vs.* DiT-XL/2-G), and better FID than the existing work with similar computation (*e.g.*, Ours *vs.* U-ViT-L/2).

| Model | $256 \times 256$ | | | $512 \times 512$ | | |
|---|---|---|---|---|---|---|
| | FLOPs | Throughput A100 (B=64) samples/sec | FID | FLOPs | Throughput A100 (B=64) samples/sec | FID |
| ADM [18] | 110G | - | 10.60 | - | - | - |
| LDM [13] | 104G | 362 | 3.60 | - | - | - |
| U-ViT-L/2 [86] | 77G | 498 | 3.40 | 340G | 86 | 4.67 |
| U-ViT-H/2 [86] | 133G | 271 | 2.29 | 546G | 45 | 4.05 |
| DiT-XL/2-G [4] | 118G | 293 | 2.27 | 525G | 51 | 3.04 |
| Ours | 52G | 556 | 2.41 | 224G | 130 | 3.28 |
| Ours (sampled cfg [96]) | 52G | 556 | 2.23 | 224G | 130 | 3.15 |

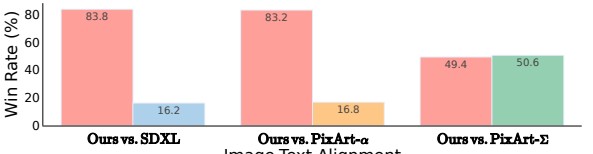

Figure 5: **Image-Text Alignment Study.** We perform user study for 1000 prompts and ask them to choose images with better image-text alignment. It shows that we outperform SDXL and PixArt-$\alpha$. While our performance is on par with PixArt-$\Sigma$, Tab. 5 shows that we yield more realistic generations.

Table 5: **Validation Data Evaluation.** We use the validation set of captioned data for computing the FID scores for comparison between different models (see details in Appendix A.5.2)

| Model | FID Set-B-10K | FID Set-A-10K |
|---|---|---|
| PixArt-$\alpha$ | 46.03 | 20.12 |
| PixArt-$\Sigma$ | 40.01 | 19.25 |
| SDXL | 35.86 | 16.49 |
| Ours | 15.45 | 10.88 |

clearly indicates that our generations align well with the real-world images. This is also evident from our earlier comparison on qualitative visualization in Fig. 4.

**Resource Efficiency.** We compare the resource requirements of our model against various baselines in Appendix Tab. 12. It shows that we achieve better inference latency compared to many existing models. Further, we consume considerably less compute to achieve much better performance than many existing baselines as illustrated by evaluations in the previous section.

## 5 Conclusion

In this work, we design hybrid architectures comprising convolutional and transformer blocks with applications to many computer vision tasks. We focus on developing a simple hybrid model with better throughput and performance trade-offs. Instead of designing efficient alternatives to the convolutional and transformer (mainly attention mechanism) blocks, we leverage existing vanilla attention along with the FusedMBConv block to design the new architecture, called AsCAN. Our main philosophy revolves around the uneven distribution of the convolutional and transformer blocks in the different stages of the network. We refer to this distribution as *asymmetric*, in the sense that it favors more convolutional blocks in the early stages with a mix of few transformer blocks, while it reverses this trend favoring more transformer blocks in the later stages with fewer convolutional blocks. We demonstrate the superiority of the proposed architecture through extensive evaluations across the image recognition task, class conditional generation, and text-to-image generation.

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

# A  Appendix

## A.1  Image Recognition Architecture Details

We describe the three AsCAN variants (tiny, base, and large) used in the experiments (see Sec. 4.1) in the Tab. 6. We follow earlier works [31] to scale the tiny variant discussed in Sec. 3.1. Note that the stem consists of two convolutional layers that downsample the input to half the spatial resolution. The classifier stage projects the final feature map to the corresponding embedding size and performs adaptive average pooling to reduce the spatial dimensions to $1 \times 1$, in order to apply a feed-forward layer that acts as classifier head on top of these pooled features. Our convolution blocks use the batch-norm as the normalization layer while the transformer blocks leverage the layer-norm as the normalization layer. We also learn the relative positional embeddings in our attention mechanism similar to earlier works [31, 32, 46].

Table 6: **Architecture Details.** We provide detailed configuration of the three variants of the AsCAN proposed in this work, namely, tiny, base, and large. As per our method section, we use FusedMBConv (C) as the convolution block and Vanilla Transformer (T) as the transformer block. We refer the reader to the Fig. 2 for their detailed description and schematic of these building blocks. We use $K$ to denote the number of classes. Note that we hide the activation and normalization layers in the blocks. We use GeLU activation and batch-normalization in the Stem and the classifier stages.

| Stage | Tiny | Base | Large |
|---|---|---|---|
| S0: Stem | Conv $3 \times 3$, Channels 64, Stride 2
Conv $3 \times 3$, Channels 64, Stride 1 | Conv $3 \times 3$, Channels 64, Stride 2
Conv $3 \times 3$, Channels 64, Stride 1 | Conv $3 \times 3$, Channels 128, Stride 2
Conv $3 \times 3$, Channels 128, Stride 1 |
| S1: Only-Conv | C, Channels 96, Stride 2
C, Channels 96, Stride 1 | C, Channels 96, Stride 2
C, Channels 96, Stride 1 | C, Channels 128, Stride 2
C, Channels 128, Stride 1 |
| S2: Mix | C, Channels 192, Stride 2
C, Channels 192, Stride 1
C, Channels 192, Stride 1
T, Channels 192, Stride 1 | (C, Channels 192, Stride 2) $\times 1$
(C, Channels 192, Stride 1) $\times 3$
(T, Channels 192, Stride 1) $\times 2$ | (C, Channels 256, Stride 2) $\times 1$
(C, Channels 256, Stride 1) $\times 3$
(T, Channels 256, Stride 1) $\times 2$ |
| S3: Mix | C, Channels 384, Stride 2
C, Channels 384, Stride 1
T, Channels 384, Stride 1
T, Channels 384, Stride 1 | (C, Channels 384, Stride 2) $\times 1$
(C, Channels 384, Stride 1) $\times 6$
(T, Channels 384, Stride 1) $\times 7$ | (C, Channels 512, Stride 2) $\times 1$
(C, Channels 512, Stride 1) $\times 6$
(T, Channels 512, Stride 1) $\times 7$ |
| S4: Mix | C, Channels 768, Stride 2
T, Channels 768, Stride 1
T, Channels 768, Stride 1
T, Channels 768, Stride 1 | C, Channels 768, Stride 2
T, Channels 768, Stride 1
T, Channels 768, Stride 1
T, Channels 768, Stride 1 | C, Channels 1024, Stride 2
T, Channels 1024, Stride 1
T, Channels 1024, Stride 1
T, Channels 1024, Stride 1 |
| S5: Classifier | Conv $1 \times 1$, Channels 512
Adaptive Avg Pool
Feed-Forward $512 \times$ K | Conv $1 \times 1$, Channels 768
Adaptive Avg Pool
Feed-Forward $768 \times$ K | Conv $1 \times 1$, Channels 1024
Adaptive Avg Pool
Feed-Forward $1024 \times$ K |

## A.2  ImageNet Classification (Training Procedure & Hyper-parameters)

We follow the same training strategy for our ImageNet experiments in Sec. 3.1 and Sec. 4.1. We report the top-1 accuracy on the single center crop image. For training ImageNet-1K models with $224 \times 224$ resolution, we use the AdamW optimizer with a peak learning rate of $3e - 3$ for 300 epochs. We use a batch size of $4096$ images during this training period. We follow a cosine schedule for decaying the learning rate to the minimum learning rate of $5e - 6$. We also perform a learning rate warm-up to avoid instabilities during the training. We follow a 20 epoch warm-up schedule with an initial learning rate of $5e - 7$ that gets warmed up to the peak learning rate.

For all our experiments, we use standard data augmentation strategies. We use RandAugment [98] with parameters $(2, 15)$, MixUp [99] with $\alpha = 0.8$, color jittering with $0.4$ as the weight, and label smoothing with $0.1$ as the smoothing parameter. We also use $0.05$ value for the weight decay regularization and perform an exponential model averaging with a decay value of $0.9999$. In addition, we adopt gradient clipping with a gradient norm of $1.0$ to avoid instabilities during the training of such large models. We also enable stochastic depth for regularization. We use the stochastic depth of $0.3/0.4/0.5$ for the three variants in our experiments.

### A.2.1  ImageNet-1K

We benchmark various existing architectures on two accelerators (NVIDIA A100 and V100 GPUs) with different batch sizes to evaluate the inference speed of a single forward pass. Additionally, we

train the three AsCANvariants with the training procedure described in earlier sections. We show the full results corresponding to the Fig. 3 in the Tab. 7. We also include inference memory consumed by various recent works in Tab. 8.

### A.2.2 Ablations on Architecture Configuration

One of the AsCAN design principles include preferring C blocks over T blocks in a stage. We demonstrated various architecture configurations using this design choice in Tab. 2. For completeness, we reverse this design choice and prefer T blocks over C blocks in a stage. We show these configurations in Tab. 9. Using **T** block in stem / early layers result in lower throughput. Further, the performance of configurations with **T** before **C** yields lower accuracy vs throughput trade-off.

### A.2.3 ImageNet-21K

We also study the effects of pre-training the AsCAN family on larger datasets such as ImageNet-21K [72]. Similar to earlier works, we pre-train our models on the ImageNet-21K dataset. We use the pre-processed version of this dataset for ease of usage [94]. Following previous works [11, 31], we train the AsCAN model for 90 epochs on the ImageNet-21K dataset and fine-tune these weights on the ImageNet-1K classification task. Similar to the ImageNet-1K experiments, we use RandAugment [98] with parameters $(2, 5)$, MixUp [99] with $\alpha = 0.2$, color jittering with $0.4$ as the weight and label smoothing with $0.01$ as the smoothing parameter. We also use $0.01$ value for the weight decay regularization. Additionally, we perform an exponential model averaging with a decay value of $0.9999$ and gradient clipping with a gradient norm of $1.0$ to avoid instabilities during training such large models. We also enable stochastic depth for regularization. We use the stochastic depth of $0.4/0.5/0.6$ for the three variants in our experiments.

We report the performance in Tab. 10, where we observe similar top-1 accuracy as the other baselines while achieving much better inference throughput. This trend is similar to the one we observed in Sec. 4.1 when these models trained only on the ImageNet-1K dataset without any pre-training on the ImageNet-21K dataset.

### A.3 ADE20K Semantic Segmentation

**Dataset Details.** ADE20K [104] is a popular scene-parsing dataset used to evaluate the semantic segmentation performance. It consists of 20K train and 2K validation images over 150 fine-grained semantic categories. Images are resized and cropped to $512 \times 512$ resolution for training.

**Training Procedure & Hyper-parameters.** We base our semantic segmentation experiments on the widely used and publicly available mmsegmentation library [105]. We use the UPerNet [106] as our semantic segmentation architecture wherein different hybrid architectures are used as the backbones to extract the spatial feature maps. We extract the spatial feature maps at stages S2, S3, and S4, and forward these feature maps to the semantic segmentation network. All three AsCAN variants (tiny, base, large) have been initialized with the weights pre-trained on the ImageNet-1K task with $224 \times 224$ resolution. Training is performed in $512 \times 512$ resolution. Similar to FasterViT [32], we follow a similar schedule for training these segmentation models with an AdamW optimizer with a learning rate of $1e - 4$, and a weight decay of $0.05$. We use a batch size of 16 on 8 A100 GPUs. We also use stochastic depth similar to ImageNet training for controlling overfitting during these experiments.

**Experimental Setup.** We train UperNet [106] as segmentation architecture on the ADE20K dataset. We use the proposed AsCAN as the backbone pre-trained on the ImageNet-1K dataset. We use the AdamW optimizer [89] with learning rate $1e - 4$ and weight decay $0.05$. Following earlier works [32, 74], we train the models for 160K iterations using the mmsegmentation library [105]. We train these networks on 8 NVIDIA A100 GPUs using 16 as the batch size. We compute the various inference statistics with $512 \times 512$ as the image resolution.

**Experimental Results.** We compare the performance of various backbones in Tab. 11. It shows that the proposed backbone achieves competitive mIoU while achieving nearly $1.5\times$ faster inference latency compared to the baselines, measured using the frames per second metrics. This fact can be observed across different scaling of the backbones. For instance, the Swin-T backbone achieves latency of 44FPS while AsCAN-T achieves 64FPS as the latency with similar mIoU. Thus, our

models achieve a similar performance-*vs.*-latency trend at the downstream semantic segmentation task as the classification benchmark.

## A.4 T2I Dataset Details

We curate our training dataset with careful filtration on diverse data sources using first-party and licensed datasets. We depict this entire process in Fig. 6. We start by collating all these data sources to create a stream of images and if available, relevant text captions. Next stage, we add a series of filters to improve the quality of the selected images (resolution, de-duplication, aesthetic score, nsfw filtering, *etc.*). Since this data mostly contains very short descriptions or almost non-existent descriptions of the image, we unify the captioning process using an image captioning model. We collect a subset of 200K data for human annotations, where we ask the annotators to add details such as the `<angle shot of the image>`, `<image background information>`, `<human attributes>`, *etc.* We use this labeled data to fine-tune a BLIP2 [107] model which is used to generate long captions similar to this format.

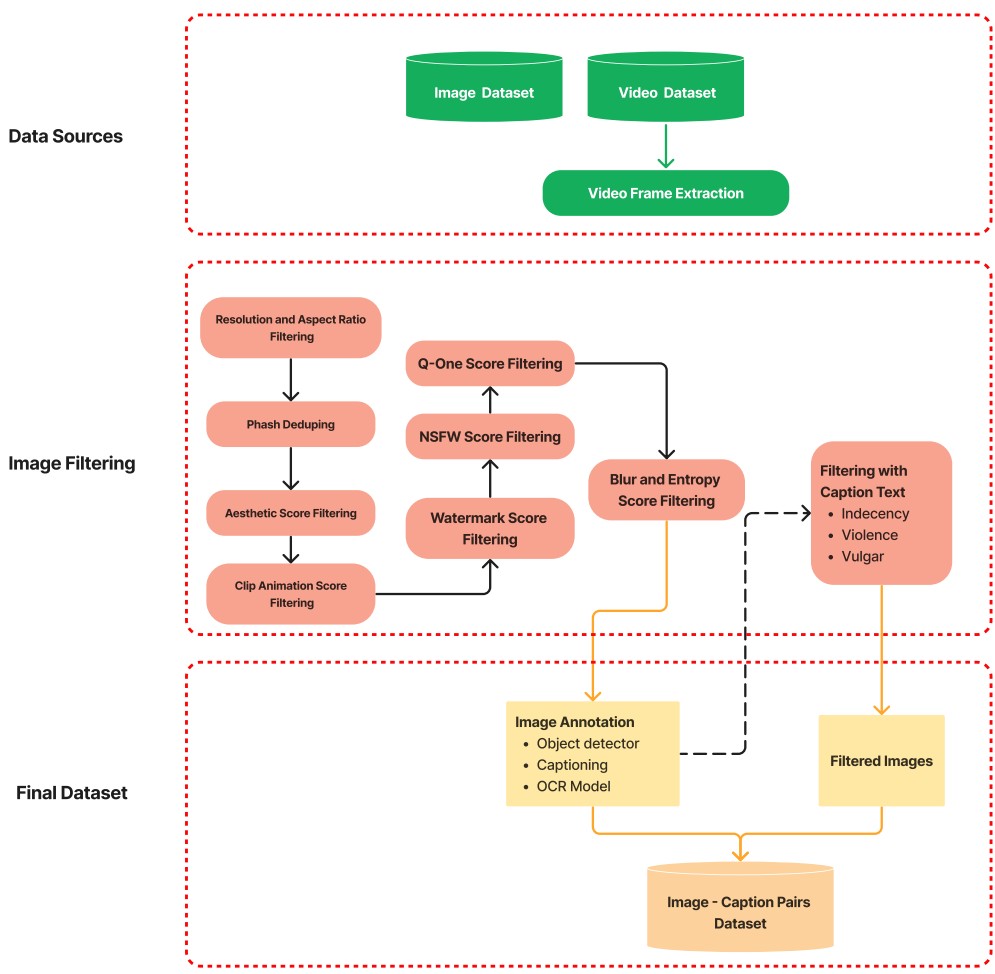

Figure 6: **Curation Pipeline for the T2I Dataset.** We show various stages involved in the creation of our T2I dataset.

### A.5    T2I Evaluation Details.

#### A.5.1    Resource Usage

We compare the resource usage of various existing methods with our proposed T2I model. Tab. 12 shows the parameter count of the model, training compute cost, and the inference cost per image generation.

#### A.5.2    Evaluation on Internal Data

We evaluate our models against various baselines on two internally captioned datasets (Set-A-10K and Set-B-10K). We draw these datasets from our validation set. We compute the FID between the generated 10K images against the reference images in Tab. 5. This table also shows that fast training pipelines like PixArt-$\alpha$ and PixArt-$\Sigma$ suffer considerably when evaluated on real-world images compared to models like SDXL and Ours. We also point out that SDXL has been trained with a considerably larger dataset compared to ours and uses much more computational resources.

#### A.6    Hyper-parameter Setup for Text-to-Image Generation

Since full-scale text-to-image generation task is a computationally expensive task, it becomes crucial that we choose appropriate hyper-parameters and initialization of the model before training it on the entire dataset. Thus, we choose a proxy task that can be run on a small scale and guide us in the suitable training configurations.

**ImageNet-1K T2I Generation Task.** We resort to the ImageNet-1K dataset since it contains a variety of objects (1000 objects) in different settings. Earlier works such as Pixart-$\alpha$, train their transformer architecture on the ImageNet-1K class-conditional generation tasks, wherein they incorporate the text condition via a cross-attention mechanism. This process requires the model to forget the class condition and adapt to the text condition. In this work, along the lines of [90], we simply convert the ImageNet-1K classification dataset to Text-to-Image dataset by adding the text corresponding to the `<class name>`, with the template "a photo of a `<class name>`". Specifically, the dataset provides various different names corresponding to each class, we pick up one at random during training to learn diverse text mappings. We evaluate the models by computing the FID between the 50K generated images and the reference images from the dataset.

**Embedding Pre-Computation.** Our T2I model performs diffusion in the latent space. To keep parity between this task and the full training, we also pre-compute the SDXL VAE embeddings of the ImageNet-1K dataset. Similarly, we pre-compute Flan-T5-XXL embeddings for the text condition. This saves training resources since loading VAE and Flan-T5-XXL encoders consumes significant GPU memory as well as non-trivial computation time. Thus, by pre-computing these embeddings, we increase the batch size available for training the UNet model.

**Architecture Details.** We use similar configuration for the `C` blocks as in the image recognition architecture. All our convolutional operators use $3 \times 3$ kernel sizes. The number of output channels in the three `Down` blocks are $\{320, 640, 1280\}$. Since the three `Up` blocks are reflections of the `Down` blocks, their output channels are in the reverse order, *i.e.*, $\{1280, 640, 320\}$. Given that we use the SDXL VAE to convert the $3-$channel RGB-image into $4-$channel latent space, our number of input channels is $4$ for the convolution operator in the first `Down` block. The number of attention heads in the three `T` blocks in the three `Down` stages are $\{5, 10, 20\}$. Since our `T` blocks are applied after `C` blocks, the number of output channels of the convolutional blocks along with the number of attention heads, determine the attention dimension for the transformer block. Finally, we compute the time-step and textual embeddings. The time-step integer values are converted to timestep embedding by first converting it into sinusoidal space and then projecting it through two linear layers. We process the textual embeddings using two `T` blocks for adapting the frozen embeddings (dimension 4096) for T2I generation with just one head and the same dimension as the frozen embeddings.

**Noise Level Ablation for Different Resolutions.** Since our training strategy involves jumping resolutions from $256 \to 512 \to 1024$, we ablate on the noise levels at each of these resolutions. We use the DDPM noise scheduler for injecting the noise and the level of noise is controlled by the $\beta_T$ hyper-parameter. We try four different noise levels $\{0.01, 0.02, 0.03, 0.04\}$, and compute the FID scores. We plot these scores in Fig. 7. We find that at resolution 256, noise level 0.01 produces the best FID score, and at remaining resolutions, noise level 0.02 works the best.

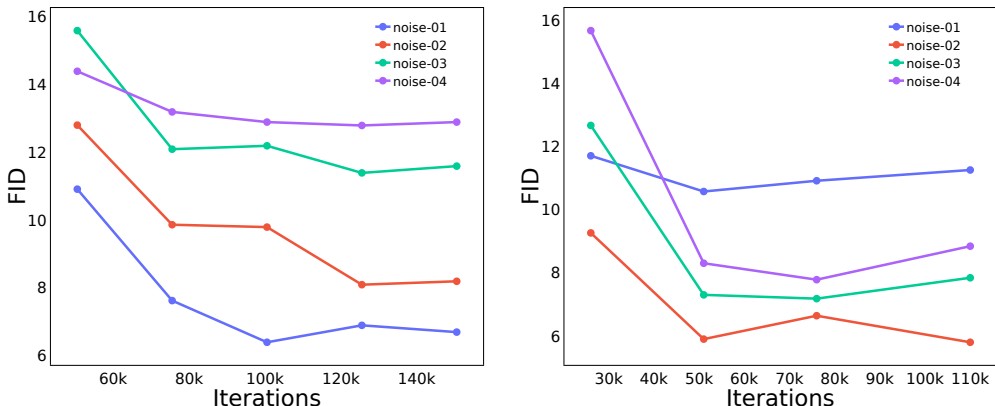

Figure 7: **Noise Level Selection for Different Resolution.** We ablate on the end noise level during the epsilon-prediction for the ImageNet-1k T2I task for the $256 \times 256$ (*Left*) and $512 \times 512$ (*Right*) resolutions with different learning rate and weight decay parameters. We plot the FID scores against a reference image set from the dataset.

**T2I Task Initialization Ablation.** We analyze initializing the $256 \times 256$ resolution training with and without pre-training from the ImageNet-1K T2I pre-trained model. As shown in Fig. 8, pre-training helps improve the performance of the model compared to training from scratch.

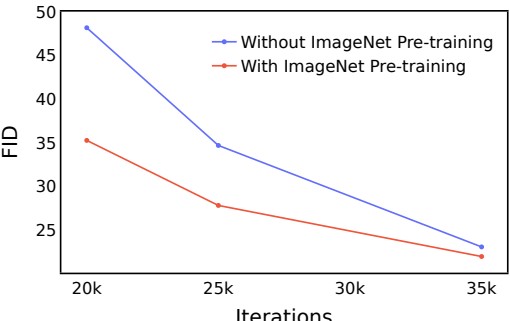

Figure 8: **Comparing T2I** $256 \times 256$ **Resolution Training with and w/o Pre-training.** We report the FID score on Set-B-10K for $256 \times 256$ resolution training, with and without pre-training from the ImageNet-1k T2I model.

### A.7    ImageNet-1K Class Conditional Generation.

We learn a class conditional image generation on the ImageNet-1K dataset with image resolution $256 \times 256$. We train a smaller variant of our asymmetric UNet architecture (as in Sec. 3.3.1) with nearly 400M parameters and inject the class condition through the cross-attention mechanism. We train this model using the DDPM scaled linear noise schedule for 1000 time steps. The training is conducted for 1000 epochs with a batch size of 2048. We use AdamW optimizer with $6e-4$ as the learning rate, 0.01 as the weight decay, and $(\beta_1, \beta_2) = (0.9, 0.99)$. We use an $8-$channel VAE to convert the input image space into a latent space for latent diffusion. We compare our results with other class conditional models in Tab. 4. Our asymmetric architecture for this task is similar to the T2I UNet architecture described earlier. To lower down the compute footprint, we reduce the number of channels to $\{160, 320, 640\}$ and reduce the cross-attention dimension from 4096 to 768. We evaluate our model with two configurations and report these two FID scores. In the first configuration, we apply a classifier-free guidance (cfg) scale [96] of 1.78 to generate the class-conditioned images. In the second configuration (sampled cfg), we apply the guidance in the steps $[5, 30]$ with a cfg scale in the increasing range $[1.1, 3.6]$. This sampling procedure is similar to the one proposed in MUSE [23]. We use the Heun [19] discrete scheduler for inference with 30 sampling steps.

### A.8 Limitations

We develop hybrid architectures with asymmetric distribution of the convolution and transformer blocks. We show that this design paradigm is generic and applicable to many vision tasks such as image recognition, semantic segmentation, and text-to-image generation. We expect this architecture design to be widely useful for other applications. From the architecture design perspective, our building blocks are not rigid and can be replaced with other efficient alternatives of the chosen C and T blocks. Currently, we focus on the vanilla attention mechanism, but we can certainly incorporate faster alternatives to quadratic attention without sacrificing any efficiency advantages. Since we demonstrate an application of the asymmetric design to the UNet framework, we list out some limitations of our text-to-image generation model, which is broadly similar to various existing text-to-image models.

- *Extra limbs.* While our model is able to generate limbs for humans and animals very well in most cases. There are instances where we can observe extra limbs in generations. It is unclear if this is due to a poorly learned diffusion process, poor captioning guidance, lack of granular information in image-caption pairs, or just an artifact of the VAE encoder while converting to the latent space.
- *Artifacts in extreme resolutions.* We train our T2I model with multi-aspect ratio bucketing. While it is able to adapt to many different resolutions in a graceful manner, we find that the model generates duplicate patterns in the extremes of the aspect ratio (significantly higher width than height or vice-versa). We believe this is due to the lack of training data in such resolutions.
- While we are much better at avoiding NSFW generations compared to baselines (due to careful curation of the training data), there might be a slight percentage of NSFW content, due to leakage in various filtration processes in the dataset curation. Thus, we incorporate an additional NFSW filter on top of the generation to remove such edge cases.

### A.9 Societal Impact

We design the asymmetric architecture and show the advantages it brings in many applications (image recognition, semantic segmentation, and text-to-image generation). Our architecture design targets a better trade-off between efficiency (training/inference latency) while achieving achieving good performance. In itself, the asymmetric design would reduce the amount of resources utilized by deep neural networks used commonly in vision tasks. The downstream applications of this architecture, especially in Text-to-Image generation need to be deployed with care. We expect the community to utilize proper care *w.r.t.* training data as well as the generated output. There needs to be special checks in place while preparing the training data for T2I tasks. We specially add various filters to remove malicious and harmful content from the dataset. Further, even generated images need to be processed with some filtration to remove similar harmful and malicious content.

### A.10 Text-to-Image Generation Results.

For visualization purposes, we generate more images with different prompts using our multi-aspect ratio model. We present these images in Fig. 9, 10, and 11. It shows that our model is able to generate a wide range of images, both very realistic as well as stylistic images. In addition, it is able to grasp various concepts and compositions.

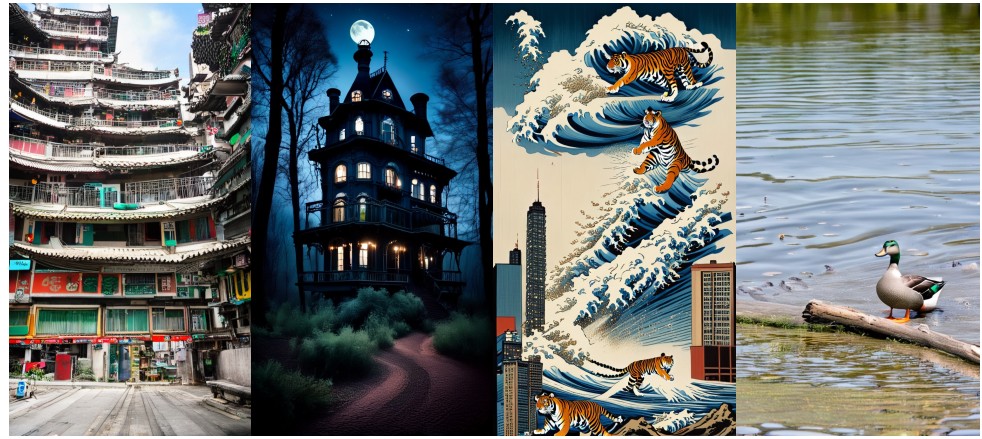

| old kowloon walled city hong kong | victorian gothic steampunk outdoor mansion and woods at night unknown creatures in the woods | A tornado made of tigers crashing into a skyscraper. painting in the style of Hokusai. | There is a large gray duck next to the water. There are small waves in the water. There is a log next to the water. |

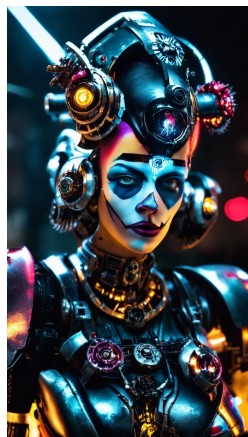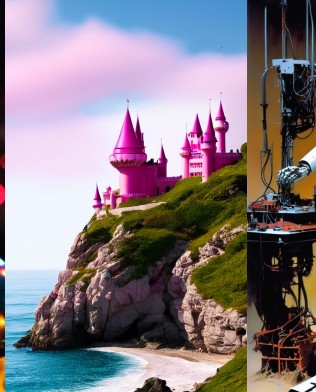

| harlequin WOMAN, crazy face epic scène , half robotic, led lights, Warhammer 40k, UHD, surreal vibes | A magical pink castle on a rocky cliff near the sea | a humanoid robot performing maintenance on itself by Frazetta | an eye-level full shot of a peacock standing sideways on the ground there is a beige stone in the background, and trees and palms are visible in the foreground, hyperrealistic extremely detailed |

Figure 9: Example text-to-image generation results.

Table 7: **Results on ImageNet-1K [72] for Classification Task.** We compare the performance of our asymmetric architectures, AsCAN, against baselines. We report the inference latency as the throughput (images per second) that is measured by inferring images with batch size as $B$ in half-precision, *i.e.*, fp16, on an A100 GPU using torch-compile and benchmark utility from timm library [100]. A similar procedure (without torch-compile) is followed to obtain the throughput on the V100 GPU.

| | Architecture | Resolution | Params | MACs | Throughput (images/s) Batch (B) A100 B=1 | A100 B=16 | A100 B=64 | V100 B=1 | V100 B=16 | Top-1 Accuracy |
|---|---|---|---|---|---|---|---|---|---|---|
| ConvNet | EfficientNet-B6 [73] | 528 | 43M | 19.0G | 88 | 529 | 589 | 27 | 205 | 84.0% |
| | EfficientNet-B7 [73] | 600 | 66M | 37.0G | 72 | 321 | 350 | 24 | 124 | 84.3% |
| | NFNet-F0 [101] | 256 | 72M | 12.4G | 199 | 2470 | 3348 | 47 | 813 | 83.6% |
| | NFNet-F1 [101] | 320 | 132M | 35.5G | 106 | 998 | 1192 | 26 | 48 | 84.7% |
| | EfficientNetV2-S [1] | 384 | 24M | 8.8G | 112 | 1884 | 2744 | 34 | 561 | 83.9% |
| | EfficientNetV2-M [1] | 480 | 55M | 24.0G | 84 | 918 | 1094 | 26 | 329 | 85.1% |
| | ConvNeXt-S [74] | 224 | 50M | 8.7G | 174 | 2291 | 2889 | 56 | 836 | 83.1% |
| | ConvNeXt-B [74] | 224 | 89M | 15.4G | 167 | 1760 | 2073 | 58 | 619 | 83.8% |
| | ConvNeXt-L [74] | 224 | 198M | 34.4G | 168 | 1045 | 1127 | 58 | 362 | 84.3% |
| | RDNet-S [102] | 224 | 50M | 8.7G | 102 | 1782 | 2761 | 53 | 780 | 83.7% |
| | RDNet-B [102] | 224 | 87M | 15.4G | 80 | 1578 | 1891 | 48 | 589 | 84.4% |
| | RDNet-L [102] | 224 | 186M | 34.7G | 78 | 640 | 990 | 32 | 290 | 84.8% |
| ViT | ViT-B/16 [10] | 384 | 86M | 55.4G | 212 | 1006 | 1266 | 104 | 86 | 77.9% |
| | ViT-B/32 [10] | 384 | 307M | 190.7G | 112 | 893 | 924 | 54 | 27 | 76.5% |
| | DeiT-B [103] | 384 | 86M | 55.4G | 189 | 1058 | 1192 | 130 | 488 | 83.1% |
| | Swin-S [46] | 224 | 50M | 8.7G | 50 | 841 | 2221 | 34 | 436 | 83.0% |
| | Swin-B [46] | 384 | 88M | 47.0G | 51 | 458 | 486 | 20 | 85 | 84.5% |
| | PVTv2-B3[91] | 224 | 45M | 7G | 49 | 933 | 3782 | 33 | 548 | 83.2% |
| | PVTv2-B5[91] | 224 | 82M | 11.8G | 32 | 511 | 2112 | 17 | 291 | 83.8% |
| | EfficientViT-B2[93] | 224 | 24M | 1.6G | 142 | 2171 | 5132 | 48 | 782 | 82.1% |
| | EfficientViT-B3[93] | 224 | 49M | 4G | 114 | 1882 | 2860 | 32 | 552 | 83.5% |
| | MOAT-2[92] | 224 | 73M | 17.2G | 132 | 1367 | 2087 | 38 | 424 | 84.7% |
| | MOAT-3[92] | 224 | 190M | 44.9G | 70 | 805 | 962 | 21 | 246 | 85.3% |
| | SMT-S[79] | 224 | 21M | 4.7G | 38 | 566 | 2211 | 43 | 348 | 83.7% |
| | SMT-B[79] | 224 | 32M | 7.7G | 27 | 388 | 1412 | 14 | 243 | 84.3% |
| | MogaNet-L [82] | 224 | 83M | 15.9G | 34 | 523 | 882 | 24 | 290 | 84.7% |
| | MogaNet-XL [82] | 224 | 181M | 34.5G | 32 | 471 | 576 | 19 | 210 | 85.1% |
| | BiFormer-S [80] | 224 | 26M | 4.5G | 45 | 937 | 2139 | 36 | 595 | 83.8% |
| | BiFormer-B [80] | 224 | 57M | 9.8G | 50 | 840 | 1439 | 28 | 440 | 84.3% |
| | RMT-S [81] | 224 | 27M | 4.5G | 46 | 790 | 2439 | 38 | 480 | 84.1% |
| Hybrid | CoAtNet-0 [11] | 224 | 25M | 4.2G | 214 | 3537 | 5221 | 61 | 976 | 81.6% |
| | CoAtNet-1 [11] | 224 | 42M | 8.4G | 141 | 2221 | 2907 | 45 | 629 | 83.3% |
| | CoAtNet-2 [11] | 224 | 75M | 15.7G | 133 | 1718 | 2040 | 38 | 540 | 84.1% |
| | CoAtNet-3 [11] | 224 | 168M | 34.7G | 132 | 1085 | 1105 | 37 | 388 | 84.5% |
| | MaxViT-T [31] | 224 | 31M | 5.6G | 73 | 1098 | 2756 | 23 | 357 | 83.62% |
| | MaxViT-S [31] | 224 | 69M | 11.7G | 70 | 1019 | 1775 | 24 | 243 | 84.45% |
| | MaxViT-B [31] | 224 | 120M | 23.4G | 34 | 507 | 1012 | 11 | 164 | 84.95% |
| | MaxViT-L [31] | 224 | 212M | 43.9G | 34 | 544 | 759 | 10 | 123 | 85.17% |
| | FasterViT-1 [32] | 224 | 53M | 5.3G | 67 | 1123 | 4106 | 23 | 363 | 83.2% |
| | FasterViT-2 [32] | 224 | 76M | 8.7G | 64 | 1112 | 4376 | 24 | 321 | 84.2% |
| | FasterViT-3 [32] | 224 | 160M | 18.2G | 46 | 831 | 3131 | 17 | 257 | 84.9% |
| | FasterViT-4 [32] | 224 | 425M | 36.6G | 50 | 800 | 1392 | 18 | 234 | 85.4% |
| Ours | AsCAN-T | 224 | 55M | 7.7G | 199 | 3224 | 4295 | 67 | 1148 | 83.44% |
| | AsCAN-B | 224 | 98M | 16.7G | 113 | 1878 | 2393 | 38 | 590 | 84.73% |
| | AsCAN-L | 224 | 173M | 30.7G | 120 | 1381 | 1617 | 40 | 440 | 85.24% |

Table 8: **Results on ImageNet-1K for Classification Task (memory consumption).** Following Tab. 7, for recent related networks, we include memory consumed during inference with 64 batch size.

| | Architecture | Resolution | Params | MACs | Throughput (images/s) Batch (B) A100 B=1 | B=16 | B=64 | V100 B=1 | B=16 | Top-1 Accuracy | Memory (GB) B=64 |
|---|---|---|---|---|---|---|---|---|---|---|---|
| ConvNet | RDNet-S [102] | 224 | 50M | 8.7G | 102 | 1782 | 2761 | 53 | 780 | 83.7% | 12.6 |
| | RDNet-B [102] | 224 | 87M | 15.4G | 80 | 1578 | 1891 | 48 | 589 | 84.4% | 17.5 |
| | RDNet-L [102] | 224 | 186M | 34.7G | 78 | 640 | 990 | 32 | 290 | 84.8% | 26.2 |
| Hybrid | MOAT-2 [92] | 224 | 73M | 17.2G | 132 | 1367 | 2087 | 38 | 424 | 84.7% | 31.8 |
| | MOAT-3 [92] | 224 | 190M | 44.9G | 70 | 805 | 962 | 21 | 246 | 85.3% | 65.2 |
| | SMT-S [79] | 224 | 21M | 4.7G | 38 | 566 | 2211 | 43 | 348 | 83.7% | 13.5 |
| | SMT-B [79] | 224 | 32M | 7.7G | 27 | 388 | 1412 | 14 | 243 | 84.3% | 22.6 |
| | MogaNet-S [82] | 224 | 25M | 5.0G | 93 | 1593 | 2455 | 47 | 740 | 83.4% | 17.1 |
| | MogaNet-L [82] | 224 | 83M | 15.9G | 34 | 523 | 882 | 24 | 290 | 84.7% | 46.6 |
| | MogaNet-XL [82] | 224 | 181M | 34.5G | 32 | 471 | 576 | 19 | 210 | 85.1% | 74.9 |
| | BiFormer-S [80] | 224 | 26M | 4.5G | 45 | 937 | 2139 | 36 | 595 | 83.8% | 18.6 |
| | BiFormer-B [80] | 224 | 57M | 9.8G | 50 | 840 | 1439 | 28 | 440 | 84.3% | 27.9 |
| | RMT-S [81] | 224 | 27M | 4.5G | 46 | 790 | 2439 | 38 | 480 | 84.1% | 13.7 |
| Ours | AsCAN-T | 224 | 55M | 7.7G | 199 | 3224 | 4295 | 67 | 1148 | 83.44% | 9.6 |
| | AsCAN-B | 224 | 98M | 16.7G | 113 | 1878 | 2393 | 38 | 590 | 84.73% | 15.2 |
| | AsCAN-L | 224 | 173M | 30.7G | 120 | 1381 | 1617 | 40 | 440 | 85.24% | 21.2 |

Table 9: **Analysis of Architecture Configuration (with T before C).** We extend Tab. 2 by ablating over the preference of C and T blocks in a stage. Using **T** block in stem / early layers result in lower throughput. Further, the performance of configurations with **T** before **C** yields lower accuracy vs throughput trade-off.

| Block Configuration | Params | Inference(images/s) A100 B=16 | B=64 | V100 B=16 | Top-1 Acc. |
|---|---|---|---|---|---|
| CC-CCCT-CCTT-CTTT (C1) | 55M | 3224 | 4295 | 1148 | 83.4% |
| CC-CCCT-CCTT-CCTT (C2) | 73M | 3217 | 4179 | 1036 | 83.2% |
| CC-CCCT-CCTT-TTTT (C3) | 41M | 3384 | 4472 | 1224 | 82.9% |
| CC-CCCT-CCCC-TTTT (C4) | 50M | 3434 | 4411 | 1182 | 83.1% |
| CC-CCCT-CCCT-CCCT (C5) | 95M | 3135 | 4066 | 991 | 82.7% |
| CC-TCCC-CCTT-CTTT (T1) | 56M | 3029 | 4021 | 945 | 82.8% |
| CC-CCCT-TTCC-CTTT (T2) | 57M | 3100 | 4092 | 1021 | 82.9% |
| CC-CCCT-CCTT-TTTC (T3) | 64M | 3190 | 4193 | 1045 | 83.1% |
| TT-CCCT-CCTT-CTTT (T4) | 55M | 1428 | 1584 | 487 | 83.1% |
| TT-TTTC-TTCC-TTTC (T5) | 100M | 1280 | 1440 | 390 | 83.5% |

Table 10: **ImageNet-21K [72] Pre-training.** We report the performance of models pre-trained on ImageNet-21K with 224 resolution and fine-tuned on ImageNet-1K dataset. We report the inference latency as the throughput (images per second) that is measured by inferring images with batch size as $B$ in half-precision, *i.e.*, fp16, on an A100 GPU using torch-compile and benchmark utility from timm library [100].

| | Architecture | Resolution | Params | MACs | Batch (B=16) Throughput (images/s) | Batch (B=64) Throughput (images/s) | Top-1 Accuracy |
|---|---|---|---|---|---|---|---|
| | ConvNeXt-L [74] | 224 | 198M | 34.4G | 1045 | 1127 | 86.6% |
| | MaxViT-L [31] | 224 | 212M | 43.9G | 544 | 759 | 86.7% |
| | FasterViT-4 [32] | 224 | 425M | 36.6G | 800 | 1392 | 86.6% |
| Ours | AsCAN-L | 224 | 173M | 30.7G | 1381 | 1617 | 86.7% |

Table 11: **Semantic Segmentation Results on ADE20K [104].** We compare different backbones on the ADE20K [104] semantic segmentation benchmark with UPerNet [106] as the detection architecture. Computational and storage statistics are computed using the input resolution of $512 \times 512$.

| Backbone | Latency (frames/s) A100 | Latency (frames/s) V100 | Params | MACs (G) | mIoU |
|---|---|---|---|---|---|
| Swin-T [46] | 44 | 23 | 60M | 237 | 44.5 |
| FasterViT-2 [32] | 47 | 25 | - | - | 47.2 |
| AsCAN-T (Ours) | 64 | 30 | 86M | 264 | 47.3 |
| Swin-S [46] | 27 | 16 | 81M | 261 | 47.7 |
| FasterViT-3 [32] | 34 | 19 | - | - | 48.7 |
| AsCAN-B (Ours) | 44 | 24 | 128M | 311 | 48.9 |
| Swin-B [46] | 23 | 13 | 121M | 301 | 48.1 |
| FasterViT-4 [32] | 28 | 15 | - | - | 49.1 |
| AsCAN-L (Ours) | 36 | 19 | 204M | 397 | 50.3 |

Table 12: **Resource Usage Comparison.** We compare the training time required to learn various diffusion models in the literature to our proposal. We also include parameter count and inference latency of these models.

| Model | Params | Train GPU A100 days | Inference per image ($512 \times 512$) | Inference per image ($1024 \times 1024$) |
|---|---|---|---|---|
| RAPHAEL | 3.0B | 60,000 | - | - |
| DALL.E 2 | 6.5B | 41,667 | - | - |
| Imagen | 3.0B | 7,132 | - | 13.3s |
| SDv1.5 | 0.9B | 6,250 | 1.9s | - |
| SDXL+Refiner | 2.6B | - | 7.9s | 12.2s |
| Wurstchen | 1.0B | 810 | 2.5s | - |
| GigaGAN | 0.9B | 4,783 | - | |
| PixArt-$\alpha$ | 0.6B | 753 | 2.1s | 7.2s |
| Ours | 2.4B | 2,688 | 2.2s | 5.5s |

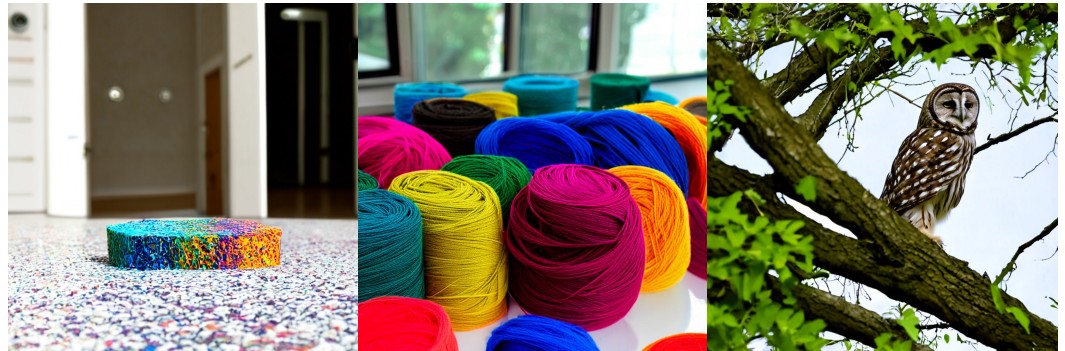

*In this image we can see an object with different colors placed on the surface and in the background there are doors and walls.*

*In this image I can see colorful threads rolls , which are on the surface. In the background it looks like a window.*

*a barred owl peeking out from dense tree branches*

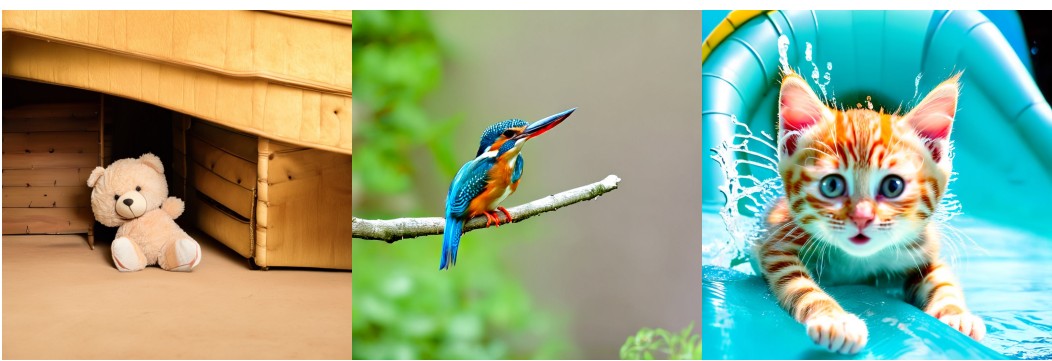

*A teddy bear under some furniture that appears to be turned on it's side.*

*A bird known for its distinctive blue and orange plumage. The kingfisher is perched on a branch, its body angled slightly to the left as if poised to take flight at any moment.*

*A cute orange kitten sliding down an aqua slide. happy excited. 16mm lens in front. we see his excitement and scared in the eye. vibrant colors. water splashing on the lens*

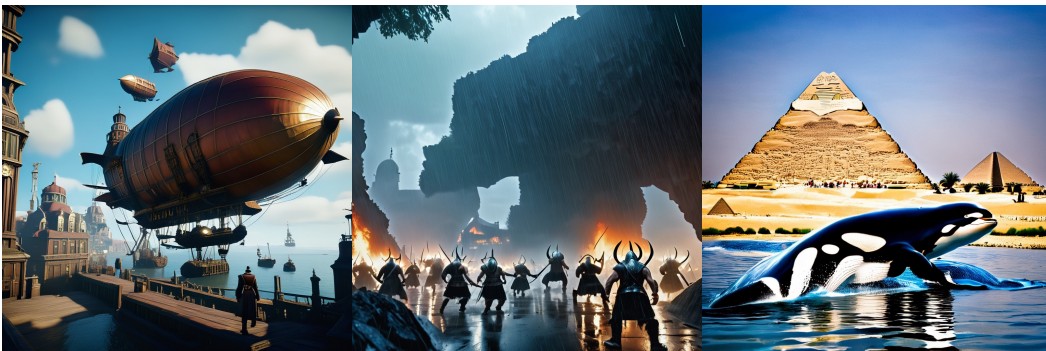

*airship docked at the port of a city in the sky by Ivan shishkin, artstation, bioshock infinite, 35mm, cinematic, --ar 2:1*

*vikings war, cave troll, viking army, viking fighting in the rain, city on fire, rain, Richard Schmid  art style, matte painting, cinematic, epic —ar 3:1*

*An orca whale swimming in the Nile River in front of an Egyptian pyramid*

Figure 10: Example text-to-image generation results.

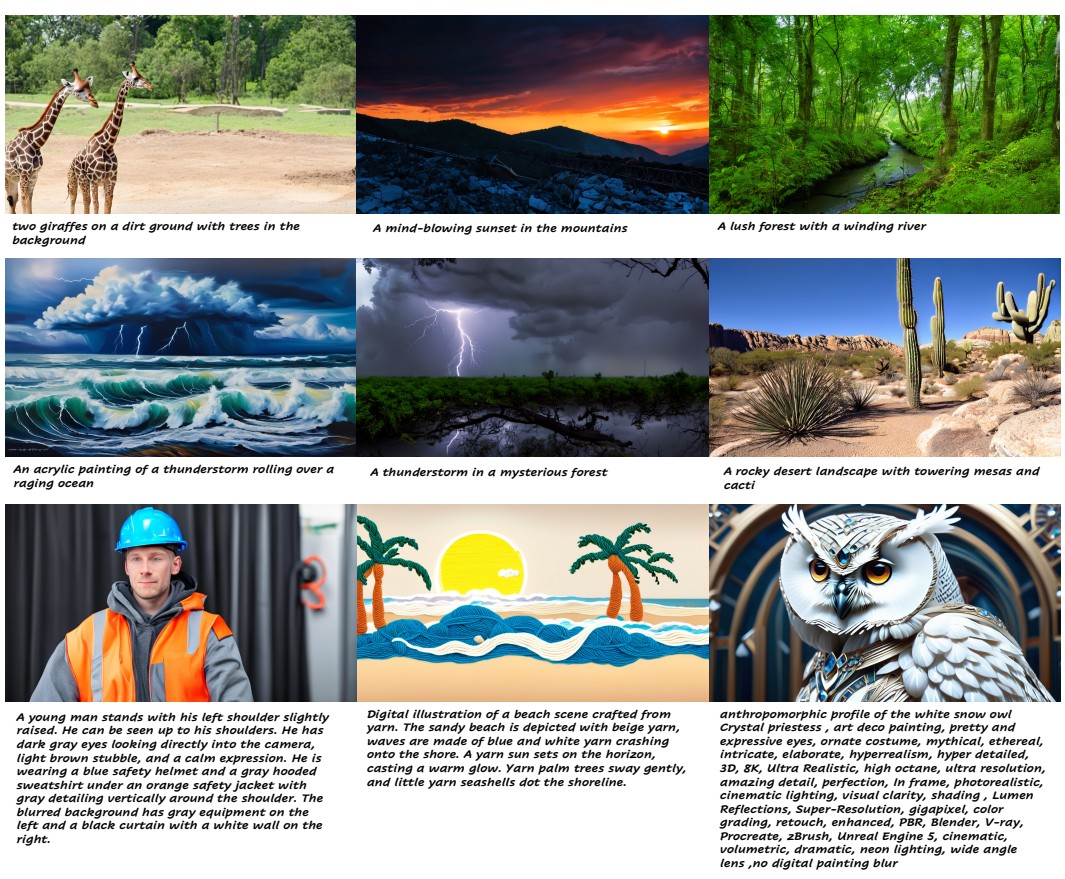

Figure 11: Example text-to-image generation results.

