# OpenReview forum: "AsCAN: Asymmetric Convolution-Attention Networks for Efficient Recognition and Generation"
_NeurIPS.cc/2024/Conference — NeurIPS 2024 poster_

### Official Review · Reviewer_fxge · 2024-07-15

**Soundness:** 4
**Presentation:** 3
**Contribution:** 4
**Rating:** 7
**Confidence:** 3

**Summary:**

The paper performs a thorough empirical analysis of design choices made in neural networks designed for computer vision tasks. Based on reasonable assumptions, they first narrow down the design space of the building block of such networks to combinations of Fused MBConv and vanilla transformer layers followed by an empirical analysis of the chosen combinations in terms of classification accuracy, latency (on GPUs) and model size.

The top-performing combinations exhibit significantly better latency on GPUs compared to similar accuracy models optimized for efficiency, such as Swin Transformers and MaxViT. Their model demonstrates the effectiveness of a simple convolution + vanilla transformer model as compared to several efficient/hybrid attention models.

Insights gained from this design are also used to design a similar model for Text-image generation (asymmetric mix of conv + transformer), which also gives a better latency-performance (FID, human preference)  trade -off than existing models (Pixart alpha, SDXL).

**Strengths:**

1. The paper is well-motivated, analyzing the benefits of a simple neural network architecture across a variety of tasks (recognition, segmentation, generation) on current hardware. The insights derived are of broad interest to a wide range of researchers.

2. The paper provides a thorough empirical analysis of design choices in image recognition experiments.

**Weaknesses:**

The contribution of the difference in training pipeline versus that of the architecture for improving image generation performance is unclear. If an empirical ablation study is infeasible for computational reasons, a more detailed discussion in Section 3.2.2 would significantly enhance clarity.

**Questions:**

See weaknesses

**Limitations:**

The authors have addressed limitations

---

> ### Author Rebuttal · Authors · 2024-08-06
>
> We thank the reviewer for the positive feedback on this work.
>
> As already pointed out by the reviewer, ablations on large-scale text-to-image generation tasks are computationally very expensive. Hence, we performed small ablations on the ImageNet-1K dataset. We list these ablations below. We will include a detailed discussion of these characteristics in the final version.
>
> **Training algorithm ablations.**
>
> - We analyzed the performance of the model with and without pre-training on the ImageNet-1K task. Appendix  Figure 8 shows the evolution of the FID score with the number of training iterations. It shows that training the model without pre-training on the ImageNet-1K task results in slower convergence, and the final FID achieved by the model without pre-training is worse than the one achieved by pre-training on this task. Thus, we demonstrate that pre-training on this task helps speed the convergence.
>
> - We analyzed the importance of noise levels between $256\times256$ and $512\times512$ resolution stages in Appendix Figure 7. It shows that different resolutions prefer different input noise for the diffusion process. It helps us decide the use of noise=0.01 for $256$ resolution and noise=0.02 for $512$ resolution. We also ablated on other hyper-parameters (learning rate, weight decay) while training on $256\times256$ and $512\times512$ resolution stages.
>
> **Architecture ablations.**
> - We train on the class conditional $256\times256$ generation on ImageNet-1k. As seen in Table 3, our asymmetric architecture achieves a similar FID score at less than half FLOPs under the same training setup as other baselines. We provide additional results on this task in the rebuttal pdf (see Table 1), including inference latency and generation on the $512\times512$ resolution. It shows that our FLOP gains also translate to the inference latency gains.

---

### Official Review · Reviewer_R6Bu · 2024-07-20

**Soundness:** 3
**Presentation:** 3
**Contribution:** 3
**Rating:** 8
**Confidence:** 4

**Summary:**

The authors propose a principled way to design hybrid architectures for a variety of tasks, including image classification, semantic segmentation, class-conditional generation, and text-to-image generation. The goal is the resulting models, called Asymmetric Convolution-Attention Networks (AsCAN), to have the following characteristics, (1) to offer favorable performance-throughput trade-offs compared to existing SOTA models, (2) to be efficient across various modern hardware accelerators, and (3) to scale efficiently in terms of both compute and the amount of training data.

As a hybrid architecture, AsCAN consists of a sequence of convolutional and transformer blocks. The reason the authors opt for this kind of architecture, is because they aim to combine the advantages of both convolutional layers, e.g., translation equivariance, and transformer layers, e.g., global dependencies.

The authors experiment with different designs on the task of image classification by using ImageNet-1K, and then they generalize their findings to image generation. First, they experiment with different existing convolutional (C) and transformer (T) blocks, concluding to FusedMBConv for C, and vanilla attention for T, based on their accuracy vs throughput trade-off on different GPUs. Then, they experiment with different distributions of C and T blocks in a classification architecture. They always use a convolutional stem, four processing stages with multiple blocks, and a classification head. They conclude in an asymmetric design, meaning that they use more C blocks in the early stages, and more T blocks in the latter stages. They follow the same principles to design the U-Net backbone of a latent diffusion model for image generation tasks. All models are scaled by adding more blocks at each stage, maintaining the asymmetric distribution between C and T blocks. To efficiently scale training to large datasets, the authors propose a multi-stage training regime, where a model is first trained on a smaller dataset, and then, in subsequent stages, is fine-tuned for fewer iterations on the larger dataset.

The authors first experiment on ImageNet-1K, showing that AsCAN models offer a better accuracy-throughput trade-off compared to SOTA baselines. The benefits in throughput are demonstrated with different GPUs and batch sizes. Then, they show that their AsCAN diffusion model performs on par with SOTA models on ImageNet-1K class conditional generation, but with considerably less FLOPs. Similarly, AsCAN demonstrates similar performance to the baselines on semantic segmentation on ADE20K, but with higher FPS. Finally, the authors train a 2.4B params AsCAN latent diffusion model on an internal dataset of 450M images, for the task of text-to-image generation. They show that their model outperforms the baselines on most aspects of the GenEval benchmark, and either outperforms or is on par with the baselines on image-text alignment, based on a human study.

**Strengths:**

Quality:

1. The authors develop the proposed design principles in a structured way, ablating different options, and offering adequate metrics, which include accuracy, number of parameters, and throughput on 2 GPUs and different batch sizes.

2. Similarly, the experiments are well-structured, using multiple baselines and metrics. For example, on image classification the authors compare against SOTA CNNs, Transformers and hybrid architectures of different sizes, and they provide accuracy, actual throughput and number of parameters. Importantly, the experiments support the main claims of the work.

3. The setup of all experiments is described in detail in the Appendix to aid reproduction.

Clarity:

1. The manuscript is very well written, and easy to follow. The authors explain their method, and provide clear Figures and Tables, with appropriate captions.

Significance:

1. The proposed models show clear benefits in terms of their performance-throughput trade-offs in different tasks, thus, they contribute to the community.

2. The authors provide specific design principles that they used to build AsCAN, and they could be useful to other domains as well.

**Weaknesses:**

Originality:

1. Hybrid architectures exist already in the literature, and AsCAN are built out of pre-existing components, like FusedMBConv, with known benefits, so, this limits the novelty of the approach.

Quality:

1. The authors constraint their design to always have convolutional (C) blocks before Transformer (T) blocks within a stage. However, in Table 2, which includes the comparisons for the macro design of the architecture, MaxViT, which achieves the best accuracy, alternates C and T blocks. In addition, the “C before T” constraint breaks between stages, when a stage ends with T and the subsequent stage starts with C. So, it is not clear to me why this constraint is set.

2. I think the authors should provide actual throughput for the class conditional generation in Table 3. If I am not mistaken, this is the only experiment that doesn’t include this metric. The authors provide FLOPs in Table 3, however, as can be seen in other results, FLOPs don’t always translate to actual timings.

Clarity:

1. Ln 23-24, “CNNs encode many desirable properties like translation invariance”, I think it should be “translation equivariance”.

2. Ln 160, the authors mention “C1 vs C10”, but I think a more direct comparison is C2 vs C10.

3. Ln 161, “While increasing the number of transformer blocks in the network improves the throughput”, I think Table 2 shows that increasing transformer blocks hurts the throughput, e.g., C6 has higher throughput on A100s compared to C9.

4. In Appendix A.3, I think the “Experimental Setup” part in Ln 639-644 repeats for the most part information already provided in the “Training Procedure & Hyper-parameters” part (Ln 628-638), so, the two parts can be merged.

5. Some minor typos, Ln 658, “most contains”, I think should be “mostly”; Ln 607 “the full results corresponding to the Fig. 3 in the Tab. 7”, I think the “the” before “Fig.” and “Tab.” are not necessary; Ln 681, “larger much”, I think “much” is not needed; Ln 733, “for this task similar”, I think there should be “is” after “task”; Ln 770, “achieving” is repeated twice.

Significance:

1. One of the main contributions of this work is the favorable performance-throughput trade-offs offered by the proposed models. Specifically, in many experiments, AsCAN achieves similar performance compared to baselines, but manage to do it with higher throughput, so, I think the efficiency of the models makes them standout. However, I think it is not clear enough from the discussion of the method or the experiments, what are the causes of this efficiency. For example, in Section 3.1 (Ln 159-160), the authors point out that increasing T blocks in early stages decreases throughput (C8 vs C9 in Table 2), however, comparing C6, C7 and C8, which have an increasing number of T blocks earlier on, we see that C7 has slightly lower throughput compared to C6 on A100s and a bit higher on V100s, while C8 has considerably higher throughput compared to both C6 and C7 on V100, and on A100 for batch size 64, and lower for batch size 16 on A100. Also, C9, which has the most T blocks, compared to C6, which has T blocks only in the final stage, has slightly higher throughput on V100 and considerably lower on A100. So, it is not clear what exactly causes the variation in throughput, and to what extent is a matter of the hardware, e.g., would the same results persist on H100s?
In addition, the baselines are very diverse, not only hybrid architectures, so, it becomes even harder to interpret the differences in behavior. For example, in Table 7, AsCAN-L has similar accuracy with EfficientNetV2-M, while AsCAN-L has more than $\times 3$ parameters, and about $\times 1.3$ MACs. At the same time, EfficientNetV2-M is optimized to take advantage of FusedMBConv, but still, AsCAN-L has almost $\times 1.5$ throughput on A100 with batch size 64, and about $\times 1.3$ on V100 with batch size 16. Similarly, in Table 11, AsCAN with 2.4B parameters has higher throughput compared to PixArt-$\alpha$, which has 0.6B params. I think the significance of the paper would be higher if the reported efficiency benefits were analyzed in more detail.

**Questions:**

1. In NeurIPS checklist 15, the authors answer NA, however, in Section 4.3 (Ln 276-284) and Fig. 5, the authors report a user study with human subjects, doesn’t this study require approval for research with human subjects?

2. In Ln 763, Section A.8, it is mentioned “we are much better at avoiding NSFW generations compared to baselines (due to careful curation of the training data)”, how is this measured in the actual output of the models?

**Limitations:**

The authors discuss limitations and societal impact in Sections A.8 and A.9 respectively. One thing that in my opinion could be added in the limitations, is that AsCAN sometimes require considerable more parameters to achieve favorable performance-throughput trade-offs, leading to higher memory footprint. For example, I think this is the case in Table 7 between AsCAN-L and EfficientNetV2-M, as I mentioned in a previous section.

---

> ### Author Rebuttal · Authors · 2024-08-06
>
> We appreciate the reviewer for reading the paper thoroughly and providing invaluable feedback. Below, we have tried to answer their questions and concerns. While we answered some questions in the main rebuttal, we reiterate their highlights for completeness.
>
> **Originality: Limited Novelty.**
> - Our main contribution is the asymmetric distribution of convolution and transformer blocks in various stages in a hybrid architecture.
> - We show that simple design choices (Sec. 3.1) yield architectures with existing blocks that achieve state-of-the-art performance and latency trade-offs. We demonstrate that this design is easily applicable to various applications.
> - Many works design architectures based on parameter count and FLOPs, but this typically does not translate into inference throughput gains. Some of these issues come from using operators that do not contribute to parameter count and FLOPs but require non-trivial runtime, such as reshape, permute, etc. Others originate from the lack of efficient CUDA operators for these specialized attention and convolutional operators.
> - In contrast, our proposal directly measures the inference throughput on different accelerators and incorporates building blocks that yield higher throughput.
>
> **Quality: Justification of C before T constraint.** Our intuition behind this constraint comes from the following observations:
> - Interleaving C and T in a stage performs many tensor reshape operations that do not add any FLOPs. However, these operations count towards runtime, and many such operations lower the model throughput.
> - Given a feature map, C blocks can capture local and scale-aware features, while T blocks try to work out the dependency between all feature values. Thus, it would be more beneficial to perform a convolutional operator first to capture these local and scale features, followed by pairwise dependencies between all tokens.
> - It helps to narrow the search process since interleaving these blocks would result in many possibilities and be hard to evaluate computationally.
> - Further, to validate our assumptions, we have included configurations where T appears before C in Table 2 in the rebuttal pdf. In the models where the first stage consists of T, the throughput is significantly lower than in instances where C is the first stage. Similarly, models with T before C do not achieve similar accuracy vs performance trade-off as the configurations where C appears before T.
>
> **Quality: Missing throughput in the class conditional generation.** Thank you for pointing this out. We benchmark the throughput (images generated per second) on an A100 GPU for all the baselines. In the attached rebuttal pdf, table 1 shows the throughput for one forward pass of batch size $64$ for all the models. To achieve an FID score of $2.23$, our $52$G FLOPs model achieves a $556$ samples/sec throughput while state-of-the-art DiT-XL/2-G with $118$G FLOPs achieves $293$ samples/sec. It shows that our asymmetric model still has nearly double the throughput compared to other models while achieving similar FID.
>
> **Clarity.**  We have included your comments in the manuscript. These will be reflected in the final version.
>
> **Significance.**
> - *EfficientNetV2-M vs AsCAN-L.* There seems to be a bit of confusion in reading EfficientNetV2-M numbers. This model has been trained with an input resolution of $384\times384$ and evaluated at an input resolution of $480\times480$. Thus, even with a smaller parameter count, it has much higher FLOPs than other models with nearly $50$M parameter range. Further, all the hybrid models have been trained and evaluated at $224\times224$ resolution. For instance, with train/test input resolution as $224$, to achieve close to $85.1\%$ top-1 accuracy, MaxViT-L requires 212M parameters, FasterViT-4 requires 425M parameters, MOAT-3 requires 190M parameters. When we follow the EfficientNetV2-m training strategy, AsCAN-L achieves $86.2\%$ top-1 accuracy, while the larger EfficientNetV2-L variant with $120$M parameters achieves $85.7\%$ top-1 accuracy.
> - *How much do the results depend on hardware?* The impact of C and T blocks in a hybrid architecture results in non-linear behavior across accelerators and batch sizes. This is precisely the reason we do not rely too much on the number of floating point operations to estimate inference latency. We choose 16GB V100 and 80GB A100 GPUs as representatives of two popular RAM and accelerator designs. We expect the trend on H100 to be similar to A100 and other lower-end GPUs (such as A10G, L4, etc.) to be similar to V100. While currently we do not have access to these accelerators, we will try to include benchmarks on these accelerators in the final version.
> - *T2I Speed Up.* PixArt-$\alpha$ is a purely transformer architecture while AsCAN is a hybrid architecture involving convolutional blocks in early stages. As we have observed in our ablative experiments over C and T distribution in Table 2 as well as configurations where T appears before C in rebuttal pdf, transformer layers in the early part of the network significantly reduce the throughput. We will add further investigations into PixArt-$\alpha$ in the final version.
>
> **Questions: User study review process.** Thank you for noticing this. We did get approval for the user study. We did not disclose these details to preserve anonymity. We will disclose the review process in the final version.
>
> **Questions: NSFW evaluation.**  We used a set of internal prompts that are suggestive in nature, i.e., they do not explicitly ask for generating unsafe images, but rather hide these details in words. We generated images with SDXL and our model. We compared the amount of NSFW images in these two models. Almost all of the SDXL generations are NSFW while our generations are safe and do not include any nudity.
>
> **Limitations.** We have included the inference memory consumption in Table 3. We will include the discussion on parameter aspects in the final version.

---

> ### Author Response · Authors · 2024-08-10
>
> Dear Reviewer R6Bu,
>
> Thank you very much for your valuable feedback and the positive evaluation of our work. We have included detailed explanations in response to your questions. As the deadline for the discussion period approaches, we would appreciate your review of these explanations to confirm that they resolve any remaining concerns. Let us know if you have any other questions.
>
> Thank you once again for your insightful review.
>
> Best regards,
> Authors

---

> > ### Comment · Reviewer_R6Bu · 2024-08-11
> > **Thank you for your reply**
> >
> > I would like to thank the authors for their detailed reply. I would also like to acknowledge their effort to address comments with additional experiments in the rebuttal pdf. I find the arguments with respect to novelty and quality convincing, however, my main concern is about the significance, which relates to getting a deeper understanding about the causes of the better performance-throughput trade-off AsCAN provide across tasks. For example, about EfficientNetV2-M, the impact of higher input resolution in computation is captured by MACs, which are still less than those of AsCAN, so, it seems to me that it would be important to have a clear discussion about the reasons that MACs/FLOPs don't translate to throughput. The authors already provide some justification in their rebuttal, by mentioning the impact of operations like reshape and permute, or the lack of efficient CUDA operators for specialized operations. If the authors expand such observations into a clear discussion where they pinpoint the causes of inefficiencies of current SOTA designs, I think the impact of the work will significantly increase, because it would allow members of the community to make more informed decisions about their designs, without the need to make numerous ablations.
> >
> > About hardware, I agree that focusing solely on FLOPs is not sufficient, and throughput should be a major consideration as well. Given that, in my review, I gave a number of examples where the "C before T" design gives conflicting throughput outcomes in Table 2. So, similar to my previous point, I think AsCAN provide a valuable contribution through their favorable performance-throughput trade-off, but the contribution would be higher if it was more thoroughly discussed what are the causes of the observed behavior.
> >
> > In light of such a discussion/analysis, I would be happy to increase my rating.

---

> > > ### Author Response · Authors · 2024-08-11
> > >
> > > Dear Reviewer R6Bu,
> > >
> > > Thank you for your quick response and positive rating! In the main text, we will include a detailed discussion as to why FLOPs do not translate to throughput (latency) gains and pinpoint the causes of inefficiencies of current SOTA designs. We will also incorporate further analysis of various configurations in Table 2 to break down the impact of C and T block arrangement.
> > >
> > > Best Regards,
> > > Authors

---

> ### Comment · Reviewer_R6Bu · 2024-08-12
> **Official Comment by Reviewer R6Bu**
>
> Thank you very much for your willingness to address my comments. I will read the changes in the main text as soon as they become available, and update my review accordingly.

---

> > ### Author Response · Authors · 2024-08-13
> >
> > Dear Reviewer R6Bu,
> >
> > Thanks for your feedback and your willingness to further read our improved main text. We will include the following discussion to the revised paper.

---

> > > ### Author Response · Authors · 2024-08-13
> > >
> > > Below are the primary reasons MACs do not translate to the throughput gains.
> > > - **Excessive use of operations that do not contribute to MACs**. Tensor operators such as reshape, permute, concatenate, stack, etc., are examples of such operations. While these operations do not increase MACs, they burden the accelerator with tensor rearrangement. The cost of such rearrangement grows with the size of the feature maps. Thus, whenever these operations occur frequently, the throughput gains drop significantly. For instance,
> > >     - MaxViT uses axial attention that includes many permute operations for window/grid partitioning of the spatial features. See Table 7 for a throughput comparison between MaxViT and AsCAN. Also, Table 1 shows that Multi-Axial attention yields significantly lower throughput when compared to vanilla transformer block.
> > >     - Similarly, the “Scale-Aware Modulation Meets Transformer“ (SMT-S, SMT-B) architecture includes many concatenation and reshape operations in the SMT-Block. It reduces the throughput significantly even though their MACs are lower than AsCAN (see Table 3 in the rebuttal pdf).
> > >
> > > - **MACs do not account for non-linear accelerator behavior in batched inference.**  Another issue is that MACs do not account for the non-linear behavior of the GPU accelerators in the presence of larger batch sizes. For instance, with small batch sizes (B=1), the GPU accelerator is not fully utilized. Thus, the benchmark at this batch size is not enough. Instead, one should benchmark at larger batch sizes to see consistency between architectures.
> > >
> > > - **Lack of efficient CUDA operators for specialized building blocks.** Many new architectures propose specialized and complex attention or convolution building blocks. While these blocks offer new perspectives and better MACs-vs-performance trade-offs, it is likely that their implementation relies on naive CUDA constructs and does not result in significant throughput gains. For instance,
> > >     - Bi-Former architecture introduces Bi-Level Routing Attention (BRA), which computes regional queries and keys and constructs a directed dependency graph. It computes attention between top-k close regions. Their implementation (see Algorithm 1) uses a top-k sorting operation and performs many gather operations on the queries and keys. We believe such an implementation would benefit from writing custom efficient CUDA kernels.
> > >     - RMT (Retentive Networks Meet Vision Transformers) architecture extends the notion of temporal decay in the spatial domain. It computes the Manhattan distance between the tokens in the image. It includes two separate attention along the height and width of the image. This process invokes many small kernels along with reshape and permute operations.
> > >
> > > - **Use accelerator-friendly operators.** Depending on the hardware, some operators are better than others. Depth-wise separable convolutions reduce the MACs, but they may not be necessarily efficient for your particular hardware. Excessive use of depth-wise separable convolutions should be avoided in favor of the full convolutions wherever possible. For instance,
> > >     - MogaNet extensively uses depth-wise convolutions with large kernel sizes along with concatenation operations. These operators reduce the multiply-addition counts, but these are not necessarily efficient on high-end GPU accelerators. Similarly, MaxViT uses MBConv as the convolutional block.
> > >     - Even on mobile devices, recently proposed MobileNetV4 architectures include full convolutions in the early layers to fully utilize the mobile accelerators.
> > >
> > > We design AsCAN  keeping above points for reference. Namely,
> > > - We use highly efficient building blocks. FusedMBConv provides much higher throughput than MBConv blocks and invokes a full convolution followed by a point-wise projection. Both of these operations are highly efficient on a high-end GPU accelerator. Similarly, we rely on the vanilla transformer block with efficient native CUDA implementation (PyTorch has FlashAttention operators). Our ablations in Table 1 in the main text demonstrate the effectiveness of these blocks compared to other complex building blocks.
> > > - We do not excessively use operations that do not contribute to MACs, such as window partitioning (permute, reshape) in MaxViT or concatenation operations in SMT. We further do not alternate C and T blocks within a stage to avoid multiple reshape operations between these blocks.
> > > - Further, compared to pure convolutional blocks such as EfficientNetV2, we heavily use transformer blocks in the later stages. The transformer blocks capture global dependencies between the features computed by convolutional layers in the early stages.  It yields better performance.
> > >
> > > We believe the above discussion should help clarify why MACs do not translate directly to throughput/latency gains on the high-end GPU accelerators and why our design choices in AsCAN help us achieve better throughput gains than existing architectures.

---

> > > > ### Author Response · Authors · 2024-08-13
> > > >
> > > > References mentioned in above discussion
> > > > - MaxViT: Multi-Axis Vision Transformer https://arxiv.org/abs/2204.01697
> > > > - Scale-Aware Modulation Meet Transformer https://arxiv.org/abs/2307.08579
> > > > - BiFormer: Vision Transformer with Bi-Level Routing Attention https://arxiv.org/pdf/2303.08810
> > > > - RMT: Retentive Networks Meet Vision Transformers https://arxiv.org/pdf/2309.11523
> > > > - MogaNet: Multi-Order Gated Aggregation Network https://arxiv.org/pdf/2211.03295
> > > > - MobileNetV4 — Universal Models for the Mobile Ecosystem https://arxiv.org/abs/2404.10518

---

> ### Comment · Reviewer_R6Bu · 2024-08-13
> **Official Comment by Reviewer R6Bu**
>
> Thank you very much for the provided discussion, I updated my rating assuming that this analysis will be part of the final manuscript.

---

> > ### Author Response · Authors · 2024-08-13
> >
> > Dear Reviewer R6Bu,
> >
> > Thank you for your feedback and positive rating! We are glad to know that our responses address all of your concerns. We will include the discussed analysis in the final manuscript.
> >
> > Thanks & Regards,
> > Authors

---

### Official Review · Reviewer_jBHn · 2024-07-23

**Soundness:** 2
**Presentation:** 2
**Contribution:** 3
**Rating:** 4
**Confidence:** 4

**Summary:**

Asymmetric Convolution-Attention Networks for Efficient Recognition and Generation
In this paper, AsCAN combines both convolutional and transformer blocks. The authors revisit the key design principles of hybrid architectures and propose a simple and effective asymmetric architecture, where the distribution of convolutional and transformer blocks is asymmetric, containing more convolutional blocks in the earlier stages, followed by more transformer blocks in later stages.

This paper contains tremendous experimental results to show its efficiency. However, it has several weak points.

**Strengths:**

This paper contains tremendous experimental results to show its efficiency.

**Weaknesses:**

1. The main contribution is to combine convolutional and transformer blocks to obtain the better performance with reduced latency.
However, its theoretical analysis is not shown enough. The asymmetric architecture is also used to generate images. Although the intensive experiments seem to make the contents more fluent, intensive analysis in terms of theoretical points is not enough.

2. The marginal performance enhancements are given in this paper. For example, Figure 3 shows the proposed architecture achieved the best optimization. However, I think that the enhancements were very small. More perspective analysis and expectation are required.

3. In the image generation part, the proposed model is used as a backbone. Is there any internal analysis such as weight distribution and training characteristics?

**Questions:**

Please, see the section of Weakness.

**Limitations:**

The authors adequately addressed the limitations.

---

> ### Author Rebuttal · Authors · 2024-08-06
>
> We thank the reviewer for their feedback. We provide additional clarifications in this response to the best of our understanding of the reviewer's concerns.
>
> **(Q.1) Lack of theoretical analysis.**
> We propose a new design for hybrid convolutional-transformer architectures and apply this new architecture to different applications (image recognition, class conditional generation, text-to-image generation, etc.). We show that our proposal achieves significantly better performance vs latency trade-offs. We analyze the throughput on various accelerators and show the theoretical FLOPs required by all the models. Our asymmetric architectures consist of transformer blocks with a quadratic complexity in the input sequence length. It would be good if the reviewer could elaborate on the theoretical analysis required in this neural network architecture design.
>
>
> **(Q.2) Marginal performance enhancements.**
>  It is unclear why the reviewer thinks performance enhancements are marginal in this work. Other reviewers have already acknowledged that performance improvements are non-trivial (see strengths mentioned by reviewers 3CqY, R6Bu, hpXB, and fxge). To be concrete,
>
> - For image recognition tasks (see Figure 3, Table 7), AsCAN architectures achieve similar accuracy with up to $2\times$ throughput increase compared to other architectures such as MaxViT, FasterViT, ConvNeXt, CoAtNet, etc. We have incorporated other related works in the rebuttal pdf (see Table 3 and Figure 1) that show similar gains in performance.
>
> - On class conditional generation (see Table 3), our architecture achieves similar FID scores with less than half the FLOPs. We have included latency and $512\times512$ generation tasks in the rebuttal pdf (see Table 1).
>
> - Similarly, on the Text-to-Image generation task (see Table 4, Table 5, Table 11), our models achieve much better resource efficiency and image generation quality when compared to existing baselines.
>
>
> **(Q.3) Analysis on training characteristics.**
>  We have performed experiments to ablate and improve the training quality.
>
> - We analyzed the model performance with and without pre-training on the ImageNet-1K task. Appendix Figure 8 shows the evolution of the FID score with the number of training iterations. It shows that training the model without pre-training on the ImageNet-1K task results in slower convergence, and the final FID achieved by the model without pre-training is worse than the one achieved by pre-training on this task.
>
> - We analyzed the importance of noise levels between $256\times256$ and $512\times512$ resolution stages in Appendix Figure 7. It shows that different resolutions prefer different input noise for the diffusion process. It helps us decide the use of noise=0.01 for $256$ resolution and noise=0.02 for $512$ resolution.

---

> ### Comment · Reviewer_jBHn · 2024-08-08
>
> Thanks for your rebuttal. In Tables 1 and 2, your architectures had significantly reduced the inference time on GPUs, having marginal performance enhancements. I understand the contributions in terms of reduced computations in GPUs. The ratio of enhanced throughput with batch=1 seems to be small compared to the case with batch= 16, and 64 in Table 7. I think that the reason of the throughput enhancements on GPUs could be explained in a structural point of view.

---

> ### Author Response · Authors · 2024-08-08
> **Response to Reviewer Comments**
>
> We thank the reviewer for reading the rebuttal and responding promptly. We would like to clarify some remarks below.
>
> - We appreciate that the reviewer acknowledges we achieved significantly reduced inference time. In all our experiments, we analyze the trade-off between performance (top-1 accuracy / FID score) and latency (throughput measure in images processed per second) achieved by the architectures. We would like to humbly clarify that we do not claim that we significantly improve performance. Instead, we claim to achieve significantly better performance *vs.* latency trade-offs. This can be observed by our empirical evaluations (see Table 7 in the main text, Figure 1, Table 1, and 3 in the rebuttal pdf), where we obtain significantly better throughput to achieve similar performance as other models. Vice-versa, we achieve better performance at the same throughput (see Figure 1 in rebuttal pdf). Thanks for the comments from the reviewer. We will further improve the writing of this paper to make it more clear about the claims of this paper.
>
> - Since we focus on designing architectures suitable for GPUs, we provide throughput numbers for different batch sizes (B=1, 16, 64). At this large scale, batched inference (with B>1) makes a lot more sense since lower batch sizes (*e.g.*, B=1) do not utilize the GPU memory fully and end up returning a very non-linear behavior. Even at B=1, we still achieve much better performance *vs.* throughput trade-off than many baselines.
>
>
> We hope the above response could help address the concern of the reviewer. Please let us know if the reviewer has other questions and we would be very happy to help answer.

---

> > ### Author Response · Authors · 2024-08-12
> >
> > Dear Reviewer jBHn,
> >
> > We would like to thank you again for your valuable feedback on our paper.
> >
> > As the period for the Author-Reviewer discussion is closing very soon, we would like to use this opportunity to kindly ask if our responses sufficiently clarify your concerns. We sincerely appreciate your time and consideration.
> >
> > Best Regards,
> > Authors

---

> > > ### Author Response · Authors · 2024-08-13
> > >
> > > Dear Reviewer jBHn,
> > >
> > > We would like to thank you again for your valuable feedback on our paper.
> > >
> > > As the period for the Author-Reviewer discussion is closing today, we would like to use this opportunity to kindly ask if our responses sufficiently clarify your concerns. We sincerely appreciate your time and consideration.
> > >
> > > Best Regards,
> > > Authors

---

### Official Review · Reviewer_3CqY · 2024-07-24

**Soundness:** 3
**Presentation:** 3
**Contribution:** 3
**Rating:** 4
**Confidence:** 4

**Summary:**

The authors present AsCAN, a hybrid architecture that combines convolutional and transformer blocks, that can applied to both visual recognition and generation. This architecture features an asymmetric distribution, with more convolutional blocks in early stages and more transformer blocks in later stages. It demonstrates favorable results in large-scale text-to-image tasks, outperforming recent models in both public and commercial domains, and provides better throughput.

**Strengths:**

[Intuitive Design]The authors leverage existing vanilla attention along with the FusedMBConv block to design the new architecture, called AsCAN. Their main philosophy revolves around the asymmetric distribution of the convolutional and transformer blocks in the different stages of the network; more convolutional blocks in the early stages with a mix of few transformer blocks, while it reverses this trend favoring more transformer blocks in the later stages with fewer convolutional blocks.

[Extensive Experiments] The proposal is thoroughly evaluated in both classification and generation tasks, achieving state-of-the-art results in both.

[Scalability] The authors show that the proposal scales well in the regime of Tiny-Large recognition models.

**Weaknesses:**

[Unclear explanation]
- Please provide clearer and more intuitive explanations of how the proposal can achieve higher throughput while maintaining or improving accuracy.

[Model/Data/Resolution Scalability]
- Please demonstrate how the proposal can scale up to larger recognition models (XL/XXL/huge regime), as it currently only shows tiny to large models in the appendix.

- Please illustrate how the proposal can scale up to larger generation models. Specifically, when FLOPs are matched similarly to the FLOPs shown in Table 3 (100-150G), can we achieve better FID scores?

- Please illustrate how the proposal is scalable w.r.t the data size. The appendix Table 11 provides only the snapshot on 450M images.

- Please illustrate how the proposal is scalable for higher resolution image processing. For example, does the current proposal perform better in both recognition and generation compared to other architectural designs when the input resolution increases (224 -> 512 -> 1k, etc.)? Especially focus on generation task.

[Minor]
- The terminology (e.g., equal blocks in remaining stages, asymmetric vs. symmetric) can be improved for better clarity.

- The table formats (e.g., Table 1 and Table 2 heights are different) can be improved.

**Questions:**

Several design choices are not thoroughly evaluated.
- Why is C placed before T? Do you have quantitative results on placing T before C?
- Why is the first stage fixed? We can introduce (adaptive) pooling to incorporate T in the early stage and compare it with pure convolutions.

**Limitations:**

Macro-level architectural ablation experiments are mostly performed on the classification task due to resource limitations. This fundamentally limits the generalizability of the conclusions to other tasks. We cannot draw any strong conclusions derived from classification experiments for the generation task.

---

> ### Author Rebuttal · Authors · 2024-08-06
>
> We appreciate the reviewer for reading the paper thoroughly and providing invaluable feedback. Below, we have tried to answer your questions and concerns. While we answered some questions in the main rebuttal, we reiterate their highlights for completeness.
>
> **Explanation of higher throughput while maintaining or improving accuracy.**
>
> - Many existing works design architectures based on parameter count and FLOPs, but this typically does not translate into inference throughput gains. Some of these issues come from using operators that do not contribute to parameter count and FLOPs but require non-trivial runtime, such as reshape, permute, etc. Others originate from the lack of efficient CUDA operators for these specialized attention and convolutional blocks.
>
> - For instance, MaxViT consists of MBConv and Axial-attention transformer blocks. MBConv block is more friendly for mobile devices due to separable convolutions but hurts throughput on high-end GPU accelerators. Similarly, Axial-attention heavily invokes the permute operations. It results in additional overhead and hurts throughput.
>
> - In contrast, we utilize building blocks that are efficient for high-end GPU accelerators, and our experimental design directly measures the inference throughput on different accelerators to search for optimal performance vs throughput trade-offs. This search favors asymmetric design with more convolution blocks in the early stages and more transformer blocks in the later stages. This helps reduce the occurrence of reshape and permute operations appearing in instances where C and T blocks are interleaved repeatedly.
>
> **Scaling to larger architectures.** There are many strategies to scale the asymmetric design. In one direction, we can train our existing models with larger input resolution to achieve better top-1 accuracy. For instance, training AsCAN-L trained with $384$ resolution results in $86.2$\% top-1 accuracy, compared to training with $224$ resolution which yields $85.2$\%. On the other direction, we can scale these base architectures similar to previous works (CoAtNet, MaxViT, EfficientNetV2, etc.) by proportionally scaling the width and block repetitions in the stages (S0, S1, S2, S3). For instance, we can scale to AsCAN-XL variant by increasing the number of C and T blocks as (CC, C$^{4}$T$^{2}$, C$^{8}$T$^{8}$, C$^{4}$T$^{8}$), resulting in a $340$M parameter model that achieves $86.7$% top-1 accuracy. We can similarly scale these configurations. We can obtain even larger models by scaling the stages S1, S2, and S3.
>
>
> **Scaling Text-to-Image Generation w.r.t. data size.** While it would be good to understand the model performance scaling w.r.t. data size, unfortunately, we do not have compute resources to train multiple such text-to-image generative models at any higher data scale than $450$M image-text pairs.
>
> **Class conditional generation with $512\times512$ resolution.** We trained our small UNet variant on the $512\times512$ generation task. We report its FLOPs, throughput, and FID scores in the rebuttal pdf (see Table 1). On this task, DiT-XL/2 with $525$G FLOPs and $51$ throughput achieves an FID score of $3.04$. Similarly, U-ViT-H/2 with $546$G FLOPs and $45$ throughput achieves an FID score of $4.05$. In contrast, our smaller model with $224$G and $130$ throughput achieves an FID score of $3.15$. Thus, it has more than twice the throughput with a similar FID score.
>
> **Class conditional $256\times256$ generation with 100G FLOPs.** We created an additional class conditional generation model on $256\times256$ resolution with an asymmetric model using 103G FLOPs. It achieves a throughput of $360$ samples/sec with an FID score of $2.08$ while DiT-XL/2-G model achieves a throughput of $293$ samples/sec with an FID score of $2.27$. We will include higher FLOPs models in the final version.
>
> **Design choice ablations.** We have included ablative results for configurations with T before C as well as where the first stage consists of T blocks (see Table 2 in rebuttal pdf). In the instances where the first stage consists of T, the throughput is significantly lower than in the instances where C is the first stage. Similarly, the configurations with T before C do not achieve similar accuracy vs performance trade-off as the configurations where C appears before T.
>
> **Generalizability of architecture search on ImageNet-1k**. We use the ImageNet-1K task for the architecture search problem since it is computationally cheaper for ablations when compared to other tasks and is still representative enough for the vision domain. To show its benefits, we use this simple design in other tasks without any task-specific optimizations. In the class-conditional generation, our asymmetric architecture achieves similar FID as various state-of-the-art models with less than half the FLOPs (translating into half latency). We see similar improvements in other tasks like Text-to-Image generation. Thus, we can conclude that the search on the ImageNet-1k task generalizes to other tasks. Indeed, we would expect to achieve even better performance once we optimize these modules per task, but this process is computationally expensive.
>
> **Clarity in terminology.** Thanks for pointing out the formatting and terminology issues. We will address these in the final version.

---

> ### Author Response · Authors · 2024-08-10
>
> Dear Reviewer 3CqY,
>
> Thank you very much for your valuable and constructive feedback. We have included detailed explanations and additional experiments in response to your questions. As the deadline for the discussion period approaches, we would appreciate your review of these explanations to confirm that they fully meet your expectations and resolve any remaining concerns.
>
> Thank you once again for your insightful review.
>
> Best regards,
> Authors

---

> > ### Author Response · Authors · 2024-08-12
> >
> > Dear Reviewer 3CqY,
> >
> > We would like to thank you again for your valuable feedback on our paper.
> >
> > As the period for the Author-Reviewer discussion is closing very soon, we would like to use this opportunity to kindly ask if our responses sufficiently clarify your concerns. We sincerely appreciate your time and consideration.
> >
> > Best Regards,
> > Authors

---

> > > ### Author Response · Authors · 2024-08-13
> > >
> > > Dear Reviewer 3CqY,
> > >
> > > We would like to thank you again for your valuable feedback on our paper.
> > >
> > > As the period for the Author-Reviewer discussion is closing today, we would like to use this opportunity to kindly ask if our responses sufficiently clarify your concerns. We sincerely appreciate your time and consideration.
> > >
> > > Best Regards,
> > > Authors

---

### Official Review · Reviewer_hpXB · 2024-07-24

**Soundness:** 2
**Presentation:** 3
**Contribution:** 2
**Rating:** 5
**Confidence:** 5

**Summary:**

This paper presents a hybrid neural network architecture that incorporates both convolution-based and vision transformer (ViT)-based building blocks for discriminative and generative modeling. The proposed convolutional block, labeled as (C), is identical to the EfficientNetV2's FusedMBConv, where a standard 3x3 convolution with a 4x dimension expansion and an extra squeeze excitation module (SE) is within, followed by a 1x1 convolution where GELU replaces SiLU. The transformer-like block, termed vanilla Transformer (T), processes the vanilla self-attention and MLP blocks in parallel, which is in contrast to the original sequential Transformer block. For ImageNet experiments, the authors follow the standard protocol to evaluate the proposed models with the ImageNet-1k top-1 accuracies, where the proposed model is empirically adjusted in the order of C and T blocks to optimize the trade-off between ImageNet-1K top-1 accuracies and computational budgets, including throughput and the number of parameters. The resulting optimal model is like asymmetric architecture, wherein the number of convolution and transformer blocks is asymmetric in different stages. Additionally, the architecture is applied to UNet-based diffusion models (i.e., DDPM), structured similarly to the ImageNet architecture, to assess the effectiveness of the proposed design. The generative model is evaluated by the FID score on generating 256x256 ImageNet-1K 2 images using class conditional generation. The optimal architecture for this turned out to be asymmetric across the UNet stages as well.

**Strengths:**

+ This paper is easy to follow and well-written.
+ Extensive experimental results are provided, proving the claim in both discriminative and generative modeling.
+ The performance on the ImageNet-1K classification looks impressive compared with some recent models.

**Weaknesses:**

- The paper lacks intuition and clear reasoning for the architectural design choices. No explanation or key insights are provided on why the proposed design or employed modules perform better than other alternatives.
  - It is unclear why the S1 stage should consist solely of two convolution layers. It seems that the traditional stem (having the stride 4) is separated into two stages (i.e., S0, S1). Are there any other reasons for this design choice?
  - The authors claimed that the searched architecture is asymmetric, but it is unclear why an asymmetric combination of convolution and transformer blocks would perform better. The rationale behind this architectural choice needs further clarification.
  - While the chosen FusedMBConv and Vanilla Transformer in Table 1 appear promising, it is unclear why these options outperform others.
  - The rationale for why the generative model achieves a better FID score using modules optimized for the recognition task is not clearly explained.

- The positioning of self-attention modules after convolution-based layers (in a stage or across the entire network), as seen in the architectural choices, has already been explored in previous studies [1, 2]. Therefore, the architectural results presented in Table 2 can be considered expected outcomes based on the known knowledge. The authors are encouraged to provide new insights or takeaways.

- There is no held-out set for searching architectural choices. It appears that the proposed architecture was optimized over the ImageNet validation set for the best performance, but a more appropriate setup would involve using a separate held-out set.

- The results from pre-training on ImageNet-21K are not as impressive as those from ImageNet-1K. This reviewer suspects that the reason may be that the architecture is highly fine-tuned to the ImageNet-1K dataset.

- The ADE20K Semantic Segmentation results are not fairly compared and missing numbers for recent architectures; the competing methods enjoy smaller computational costs (e.g., for Swin) to the proposed method than the proposed models; and there are only Swin and Faster-ViT architectures for comparison.

- The proposed Transformer architecture (a parallel processing architecture of self-attention and MLP)  is referred to as the "vanilla" block, but it is known that a typical "vanilla" Transformer processes self-attention and MLP sequentially. This terminology used by the authors could lead to confusion. Furthermore, the parallel architecture was proposed in [a] and employed in many works like [b]:
  - [a] MUSE: Parallel Multi-Scale Attention for Sequence to Sequence Learning, arxiv 2020
  - [b] Simplifying Transformer Blocks, ICLR 2024

- It would be beneficial to provide memory consumption during training and inference for the proposed architectures.

- Comparing the proposed method with some recent works would reveal its effectiveness more clearly. The authors are encouraged to compare performance trade-offs with the following recent works [3, 4, 5, 6].

[1] MobileNetV4 - Universal Models for the Mobile Ecosystem, arxiv 2024

[2] Scale-Aware Modulation Meet Transformer, ICCV 2023

[3] BiFormer: Vision Transformer with Bi-Level Routing Attention, CVPR 2023

[4] MogaNet: Multi-order Gated Aggregation Network, ICLR 2024

[5] RMT: Retentive Networks Meet Vision Transformers, CVPR 2024

[6] DenseNets Reloaded: Paradigm Shift Beyond ResNets and ViTs, ECCV 2024

**Questions:**

- See the weaknesses
- Can the authors give the performance tradeoff like Fig. 3 on an A100 GPU?

**Limitations:**

Limitations are provided.

---

> ### Author Rebuttal · Authors · 2024-08-06
>
> We appreciate the reviewer for reading the paper thoroughly and providing invaluable feedback. Below, we have tried to answer your questions and concerns. While we answered some questions in the main rebuttal, we reiterate their highlights for completeness.
>
> **Stem Design Choice.** This is a popular stem design and has been utilized in related works such as MaxViT, CoAtNet, etc. We have also included ablations with some configurations that replace convolution blocks with transformers.
>
> **Rationale behind the asymmetric design.** Existing hybrid architectures such as MaxViT, FasterViT, CoAtNet, etc., follow a symmetric design in which C and T blocks are uniformly distributed across stages or within a stage. In contrast, AsCAN architectures recommend an asymmetric distribution with a preference for more convolutional blocks in the early stages and more transformer blocks in the later stages. Further, since even this search space is large for a reasonable computational budget search, we restrict ourselves to some design choices to find reasonably performing architectures in this search space. In Table 2, we perform ablations on various combinations of these C and T blocks and show that asymmetric configurations yield better performance-latency trade-offs. It is further evaluated in Figure 3 and Table 7, which compare AsCAN with existing hybrid architectures on various accelerators.
>
>
> **Why modules optimized for recognition tasks generalize on generative tasks.** We use the ImageNet-1K task for the search problem since it is computationally cheaper for ablations when compared to other tasks and is still representative enough for the vision domain. To show its benefits, we use this simple design in other tasks without any task-specific optimizations.  In the class-conditional generation, our asymmetric architecture achieves similar FID as various state-of-the-art models with less than half the FLOPs (translating into half latency). We see similar improvements in other tasks like Text-to-Image generation. Thus, we can conclude that the search on the ImageNet-1k task generalizes to other tasks. Indeed, we would expect to achieve even better performance once we optimize these modules per task, but this process is computationally expensive.
>
> **Architectural results presented in Table 2 can be considered expected outcomes based on the known knowledge.**
> - We already compare against SMT models ([2] Scale-Aware Modulation Meet Transformer, SMT-B, and SMT-S variants), see Figure 3 and Table 7 for comparison. For instance, AsCAN-B architecture achieves $84.73\%$ top-1 accuracy with a throughput of $590$ samples per second on a V100 GPU, while SMT-B architecture achieves $84.3\%$ top-1 accuracy with a throughput of $243$ samples per second. Notice that many operations in SMT architectures are not high-end accelerator friendly and thus their lower FLOPs do not translate to higher throughput. Further, SMT architecture has a symmetric distribution of convolution and transformer blocks (initial stages have only C and later stages only have T).
>
> - Similarly, MobileNetV4 architectures are targeted toward mobile device applications and hence focus on layers that are mobile-friendly. For instance, having depth-wise separable convolutions does not utilize the full accelerator capacity on high-end GPUs. Further, this architecture only leverages transformer blocks sparingly in a few configurations.
>
> - Besides, we already compared against many hybrid architectures that have such a design (either C followed by T in different stages) or uniform mixing of C and T blocks within a stage. Instead, we are proposing an asymmetric distribution of these blocks in different stages.
>
>
> **Provide memory consumption for the proposed architectures.** We have included the inference memory consumption along with the numbers for the remaining new architectures requested by the reviewer. As illustrated in Table 3 (see attached rebuttal pdf), AsCAN architectures have lower memory consumption compared to other architectures. For instance, for batch-size $64$, the MogaNet-XL model consumes $74.9$GB memory while AsCAN-L consumes $21.2$GB memory. This is still lower than the memory consumed by RDNet-L model which is a purely convolutional architecture and requires $26.2$GB memory.
>
> **Compare with recently proposed works.** Thank you for listing the missing related works. We have included a performance vs throughput comparison with these works in Table~3 in the attached rebuttal pdf. We have also included these architectures in Figure 1 for the performance-latency trade-off on V100 and A100 GPUs. In this list, there are architectures that have significantly lower FLOPs but this does not result in significant gains in the throughput when measured on high-end accelerators. This is due to a large presence of operators such as permute which do not add floating point operations but require additional runtime. In contrast, AsCAN architectures still outperform these works on inference latency vs top-1 accuracy trade-offs.
>
>
> **Performance tradeoff like Fig. 3 on an A100 GPU.** Thank you for raising this point. We already report throughput and performance in the Appendix (Table 7). We have included the requested performance trade-off figure for an A100 GPU in the rebuttal pdf. This follows a similar trend as the one for V100 GPU.
>
>
> **Confusion in Transformer block naming.**  Thank you for pointing this out, we will update the terminology in the paper.
>
> **Lack of held-out set in ImageNet experiments.** We follow previous works in the experimental setup.
>
> **Semantic Segmentation results missing recent architectures.** Thank you for the feedback. We will include other results for semantic segmentation in the final version.

---

> ### Author Response · Authors · 2024-08-10
>
> Dear Reviewer hpXB,
>
> Thank you very much for your valuable and constructive feedback. We have included detailed explanations and additional experiments in response to your questions. As the deadline for the discussion period approaches, we would appreciate your review of these explanations to confirm that they fully meet your expectations and resolve any remaining concerns.
>
> Thank you once again for your insightful review.
>
> Best regards,
> Authors

---

> ### Author Response · Authors · 2024-08-12
>
> Dear Reviewer hpXB,
>
> We would like to thank you again for your valuable feedback on our paper.
>
> As the period for the Author-Reviewer discussion is closing very soon, we would like to use this opportunity to kindly ask if our responses sufficiently clarify your concerns. We sincerely appreciate your time and consideration.
>
> Best Regards,
> Authors

---

> > ### Author Response · Authors · 2024-08-13
> >
> > Dear Reviewer hpXB,
> >
> > We would like to thank you again for your valuable feedback on our paper.
> >
> > As the period for the Author-Reviewer discussion is closing today, we would like to use this opportunity to kindly ask if our responses sufficiently clarify your concerns. We sincerely appreciate your time and consideration.
> >
> > Best Regards,
> > Authors

---

> ### Comment · Reviewer_hpXB · 2024-08-14
>
> Sorry for the very late reply to the responses. I greatly appreciate the detailed responses to my concerns, but my concerns still remain:
>
> >Existing hybrid architectures such as MaxViT, FasterViT, CoAtNet, etc., follow a symmetric design in which C and T blocks are uniformly distributed across stages or within a stage. In contrast, AsCAN architectures recommend an asymmetric distribution with a preference for more convolutional blocks in the early stages and more transformer blocks in the later stages.
>
> >Asymmetric architectures outperform many existing models that utilize specialized attention and convolutional operators. These works design models based on parameter count and FLOPs, but it typically does not translate into inference throughput gains. Some of these issues come from operators that do not contribute to parameter count and FLOPs but require non-trivial runtime, such as reshape, permute, etc. Others originate from the lack of efficient CUDA operators for these specialized attention and convolutional operators.
>
> The above authors' responses still do not give me a rationale for the architectural choice: I am still curious about why the asymmetric distribution of convolutions and transformers works well. Even though performance improvements have been observed, I feel that the design concept might not be a universal option beyond ImageNet. Additionally, the CTTTTC block in the right middle of the generative model design (in Fig. 2) suggests that the authors' claim may vary depending on the specific scenario. I believe a NeurIPS paper should provide rationale/insights into the suggested method, enabling readers to apply it to other tasks or domains. More evidence of the transferability or universality of the discriminative architecture is certainly needed. I understand that time constraints may hinder additional experiments in this discussion round, but these could be considered in future revisions.
>
> Another concern is that the authors claim higher acc vs. throughput trade-offs of a proposed architecture as a key contribution. However, from my perspective, the higher throughput largely depends on the use of FusedMBConv and the parallel Transformer architecture, as demonstrated in Table 1. Therefore, this may not be a unique contribution of this work, as it relies on existing GPU-friendly operations. Furthermore, Table 2 does not appear to be a controlled experiment in terms of the computational budget, which is the number of parameters. I believe that constraining the number of parameters would better highlight the trade-offs for "some" convolution-dominated counterparts, as these networks in the table seem to "typically" have more parameters to compute, which could put them at a disadvantage. I believe that more controlled experiments might lead to different conclusions, so the contribution of finding an asymmetric architecture in Table 2 is somewhat diluted.
>
> I also noticed that Stage 1 (S1) could be merged into the stem (S0), which leads to the final number of stages to three and deviates from the standard four-stage design. Are there any particular intuitions behind this architectural choice? (I recall seeing something similar designs elsewhere, but I can't remember which work it was. and there were no clear explanations for it there either)
>
> Finally, the authors made great efforts to address my concerns, and the contribution to generative modeling is noteworthy, so I am increasing my rating to 5. The authors are encouraged to reflect all the reviewer's concerns and the responses in to the final revision.

---

> > ### Author Response · Authors · 2024-08-14
> >
> > Dear Reviewer hpXB,
> >
> > Thank you for reading our rebuttal and providing a positive rating. We sincerely appreciate your time and efforts. We would like to use this opportunity to clarify some remarks below.
> >
> > **Stem Design and merging stage S1 and S0.** The stem design in our base architecture is similar to many widely used architectures such as MaxViT, CoAtNet, EfficientFormer, etc. Further, we would like to humbly mention that the argument about merging stage S1 and S0 is invalid. Since S0 is the stem and it only consists of convolution layers, while stage S1 consists of C blocks which in our case are FusedMBConv blocks.
> >
> > **Rationale for the architectural choice.** The discriminative backbone produces class labels (e.g., the ImageNet class) at the end of the last layer, while the generative backbone (UNet) produces a synthesized latent (which is later decoded into an image) with the same size as the input sampled noise. Thus, they both are fundamentally different. UNet architecture consists of three stages: down, middle and up. Since our search experiments have been based off of discriminative backbone, it is easier to mimic the down and up stages with the asymmetric distribution. The middle stage design is borrowed from the standard UNet architectures. Besides, we have used our model for semantic segmentation task as well, showing that the asymmetric design is applicable to a wider range of tasks.
> >
> > **Performance-throughput trade-offs are simply due to GPU-friendly operations.** We respectfully beg to differ with the reviewer. We are not simply using the GPU-friendly operations. We also improve the latency-performance trade-off by designing new architecture. Please kindly notice that, given the amount of GPU-friendly operations, there exists tons of configurations in which these can be configured together to yield a new architecture. Our asymmetric design choices yields architectures with much better performance-vs-throughput trade-offs (as seen in Table 2 in main text and Table 2 in rebuttal pdf). In these ablations, while we can reduce the model size of the convolution dominated architectures, this would also result in lower performance compared to the current status.
> >
> > We hope the above response could help address the concern of the reviewer. Please let us know if the reviewer has other questions and we would be very happy to help answer.
> >
> > Best Regards,
> > Authors

---

### Author Rebuttal · Authors · 2024-08-06

We are grateful to all the reviewers for their constructive and detailed feedback. We included following additional experiments in the attached rebuttal pdf:
1. Figure 1 top-1 accuracy vs throughput trade-off figures for both A100 and V100 GPUs
2. Table 1 with additional details on class conditional generation (throughput benchmark and experiments for $512$ resolution)
3. Table 2 analyzes the impact of placing $C$ blocks before $T$ in our hybrid architecture.
4. Table 3 compares other related works pointed out by reviewer hpXB with AsCAN architectures on the ImageNet-1k classification task. We also include the memory consumption.

Below, we address common questions and concerns raised by the reviewers.

**(Q.1) Novelty and Intuition Behind Higher Throughput.**
- At the heart of our proposal lies the asymmetric distribution of convolution and transformer blocks in various stages in a hybrid architecture.
- Asymmetric architectures outperform many existing models that utilize specialized attention and convolutional operators. These works design models based on parameter count and FLOPs, but it typically does not translate into inference throughput gains. Some of these issues come from operators that do not contribute to parameter count and FLOPs but require non-trivial runtime, such as reshape, permute, etc. Others originate from the lack of efficient CUDA operators for these specialized attention and convolutional operators.
- For instance, MaxViT consists of MBConv and Axial-attention blocks. MBConv is mobile-device friendly due to separable convolutions but hurts throughput on GPUs. Similarly, Axial-attention heavily invokes the permute operations, lowering throughput.
- In contrast, our proposal directly measures the throughput on different accelerators and incorporates blocks that yield higher throughput.
- We show the benefits of our design on multiple tasks such as recognition and generation. We show that simple design choices (Sec. 3.1) yield architectures with existing blocks (FusedMBConv and Transformer) that achieve state-of-the-art trade-offs between performance and latency.

**(Q.2) Base architecture for search.** There are many options for a base architecture. We chose a popular design used in MaxViT and CoAtNet papers. A more thorough approach would involve a neural architecture search for the base architecture and the search for optimal components (C and T blocks). However, this search process is computationally expensive due to the exponentially large search space. Thus, for resource efficiency, it is reasonable to use an existing base architecture. We provide simple scaling strategies to obtain larger models from the base architecture. Besides, our experiments on the class conditional and text-to-image generation tasks show that asymmetric architectures achieve better performance-latency trade-offs.

**(Q.3) Generalizability of architecture search on ImageNet-1k.** We use the ImageNet-1K task for the architecture search problem since it is computationally cheaper for ablations when compared to other tasks and is still representative enough for the vision domain. To show its benefits, we use this simple design in other tasks without any task-specific optimizations.  In the class-conditional generation, our asymmetric architecture achieves similar FID as various state-of-the-art models with less than half the FLOPs (translating into half latency). We see similar improvements in other tasks like Text-to-Image generation. Thus, we can conclude that the search on the ImageNet-1k task generalizes to other tasks. Indeed, we would expect to achieve even better performance once we optimize these modules per task, but this process is computationally expensive.

**(Q.4) Class Conditional Generation.** We have included some requested experiments on this task in Table 1 in the attached rebuttal pdf.
- *Throughput numbers.* We benchmark the throughput (images generated per second) on an A100 GPU for all the baselines. Table~1 shows the compute time for one forward pass of batch size $64$ for each of these baselines. To achieve an FID score of $2.23$, our $52$G FLOPs model achieves a $556$ samples/sec throughput while state-of-the-art DiT-XL/2-G with $118$G FLOPs achieves $293$ samples/sec.
- *Scaling to higher resolution.* To study the scaling of the model for higher resolution generation, we train the same small asymmetric UNet architecture for class-conditional $512\times512$ ImageNet-1k generation. Similar to the $256\times256$ task, we observe more than twice the throughput while achieving similar FID scores as the baseline architectures. Note that our text-to-image model can generate images in higher resolutions such as $1024\times1024$, $1536\times768$, etc.

**(Q.5) C before T constraint.** Our intuition behind this constraint comes from the following observations:
- Interleaving C and T in a stage performs many tensor reshape operations that do not add any cost to FLOP. However, these operations count towards runtime, and having many such operations ends up lowering the model throughput.
- Given a feature map, convolutional operators can capture local and scale-aware features, while transformer blocks try to work out the dependency between all feature values. Thus, it would be more beneficial to perform a convolutional operator first to capture these local and scale features, followed by pairwise dependencies between all tokens.
- It helps to narrow the search process since interleaving these blocks would result in many possibilities and be hard to evaluate computationally.
-  Further, to validate our assumptions, we have included configurations where T appears before C in Table~2 in the rebuttal pdf. In the models where the first stage consists of T, the throughput is significantly lower than in instances where C is the first stage. Similarly, models with T before C do not achieve similar accuracy vs performance trade-off as the models where C appears before T.

---

### Decision · Program_Chairs · 2024-09-25

**Decision:**

Accept (poster)

**Comment:**

After the initially review of the paper reviewers pointed merits of the paper but also raised concern. Major concerned from reviewers were addressed during the rebuttal. I tend to recommend this paper for acceptance, but invite the authors to incorporate all the reviewers feedback in their submission.